# Risk Bounds of Multi-Pass SGD for Least Squares in the Interpolation Regime

**Difan Zou**\*
The University of Hong Kong
dzou@cs.hku.hk

**Jingfeng Wu**\*
Johns Hopkins University
uuujf@jhu.edu

**Vladimir Braverman**
Johns Hopkins University
vova@cs.jhu.edu

**Quanquan Gu**
University of California, Los Angeles
qgu@cs.ucla.edu

**Sham M. Kakade**
Harvard University
sham@seas.harvard.edu

## Abstract

Stochastic gradient descent (SGD) has achieved great success due to its superior performance in both optimization and generalization. Most of existing generalization analyses are made for single-pass SGD, which is a less practical variant compared to the commonly-used multi-pass SGD. Besides, theoretical analyses for multi-pass SGD often concern a worst-case instance in a class of problems, which may be pessimistic to explain the superior generalization ability for some particular problem instance. The goal of this paper is to provide an instance-dependent and algorithm-dependent excess risk bound of multi-pass SGD for least squares in the interpolation regime, which is expressed as a function of the iteration number, stepsize, and data covariance. We show that the excess risk of SGD can be exactly decomposed into the excess risk of GD and a positive fluctuation error, suggesting that SGD always performs worse, instance-wisely, than GD, in generalization. On the other hand, we show that although SGD needs more iterations than GD to achieve the same level of excess risk, it saves the number of stochastic gradient evaluations, and therefore is preferable in terms of computational time.

## 1 Introduction

*Stochastic gradient descent* (SGD) is one of the workhorses in modern machine learning due to its efficiency and scalability in training and good ability in generalization to unseen test data. From the optimization perspective, the efficiency of SGD is well understood. For example, to achieve the same level of optimization error, SGD saves the number of gradient computations compared to its deterministic counterpart, i.e., batched *gradient descent* (GD) [7, 8], and therefore saves the total amount of running time. However, the generalization ability (e.g., excess risk bounds) of SGD is far less clear, especially from theoretical perspective.

*Single-pass* SGD, a less practical SGD variant where each training data is used only once, has been extensively studied in theory. In particular, a series of works establishes excess risk bounds of single-pass SGD for learning general smooth and convex objectives [32, 24, 26] as well as learning least squares [3, 10, 18, 19, 27, 13, 42, 38]. In practice, though, one often runs SGD with *multiple passes* over the training data and outputs the final iterate, which is referred to as *multi-pass SGD* (or simply SGD in the rest of this paper when there is no confusion). Compared to single-pass SGD that has limited number of optimization steps, multi-pass SGD allows the algorithm to perform arbitrary

---

\*Equal Contribution

36th Conference on Neural Information Processing Systems (NeurIPS 2022).

number of optimization steps, which is more powerful in optimizing the empirical risk and thus leads to smaller bias error [28].

Despite the extensive application of multi-pass SGD in practice, there are only a few theoretical techniques being developed to study the generalization of multi-pass SGD. One is based on the *uniform stability* [12, 16], which is defined as the change of the model outputs under a small change in the training data. However, the stability based generalization bound is a worst-case guarantee, which is relatively crude and does not show difference between GD and SGD (See, e.g., Chen et al. [9] showed GD and SGD have the same stability parameter in the convex smooth setting). On the contrary, one easily observes a generalization difference between SGD and GD even in learning the simplest least square problem (see Figure 1). In addition, Lin and Rosasco [22], Pillaud-Vivien et al. [28], Mücke et al. [25] explored the risk bounds for multi-pass SGD using the *operator methods* that are originally developed for analyzing single-pass SGD. Their bounds are sharp in the minimax sense for a class of least square problems that satisfy certain *source condition* (which restricts the norm of the optimal parameter) and *capacity condition* (or effective dimension, which restricts the spectrum of the data covariance matrix). Still, their bounds are uniform for a class of problem instances, and cannot point-wisely adapt to each problem instance (e.g., a least square problem with a particular data covariance matrix). In particular their reults are pessimistic for the benign-overfitting [4] least square instances (see Theorem 4.2 and related discussions).

In this paper, our goal is to establish an algorithm-dependent and problem-dependent excess risk bound of multi-pass SGD for least squares. Our focus is the *interpolation regime* where the training data can be perfectly fitted by a linear interpolator (which holds almost surely when the number of parameter $d$ exceeds the number of training data $n$). We assume the data has a sub-Gaussian tail [4]. Our main contributions are summarized as follows:

- We show that for any iteration number and stepsize, the excess risk of SGD can be exactly decomposed into the excess risk of GD (with the same stepsize and iteration number) and the so-called *fluctuation error*, which is attributed to the accumulative variance of stochastic gradients in all iterations. This suggests that GD (with optimally tuned hyperparameters) always achieves a smaller excess risk than SGD for least square problems.
- We further establish problem-dependent bounds for the excess risk of GD and the fluctuation error, stated as a function of the eigenspectrum of the data covariance, iteration number, training sample size, and stepsize. Compared to the bounds proved in prior works [22, 28, 25], our bounds allow a wider range of iteration numbers $t$, and correctly vanishes when $t \to \infty$ in the benign overfitting regime [4]. In contrast, the prior results do not allow $t \to \infty$.
- We develop a new suite of proof techniques for analyzing the excess risk of multi-pass SGD. The key idea is considering the error in its matrix form and how it is updated based on the tensor operators defined by the second-order and fourth-order moments of the empirical data distribution (i.e., sampling with replacement from the training dataset), rather than the operators used in the single-pass SGD analysis that are defined based on the population data distribution [18, 42], together with a sharp characterization on the properties of the operators.

Based on the excess risk upper bounds for SGD and GD, we make the following complexity comparison between SGD and GD: to achieve the same order of excess risk, while SGD may need more iterations than GD, it can have fewer stochastic gradient evaluations than GD. For example, consider the case that the data covariance matrix has a polynomially decaying spectrum with rate $i^{-(1+r)}$, where $r > 0$ is an absolute constant. In order to achieve the same order of excess risk, we have the following comparison in terms of iteration complexity and gradient complexity[2]:

- *Iteration Complexity:* SGD needs to take $\widetilde{\mathcal{O}}(n^{\max\{0.5, \frac{r}{r+1}\}})$ more iterations than GD, with optimally tuned iteration number and stepsize.
- *Gradient Complexity:* SGD needs $\widetilde{\mathcal{O}}(n^{\max\{0.5, \frac{1}{r+1}\}})$ less stochastic gradient evaluations than GD.

**Notation.** For $n > 0$, we use $\mathrm{poly}(n)$ to define some positive high-degree polynomial functions of $n$. For two positive-value functions $f(x)$ and $g(x)$ we write $f(x) \lesssim g(x)$ if $f(x) \leq cg(x)$ for some constant $c > 0$, we write $f(x) \gtrsim g(x)$ if $g(x) \lesssim f(x)$, and $f(x) \asymp g(x)$ if both $f(x) \lesssim g(x)$ and $g(x) \lesssim f(x)$ hold. We use $\widetilde{\mathcal{O}}(\cdot)$ to hide some polylogarithmic factors in the standard big-$\mathcal{O}$ notation. For two matrices $\mathbf{A}$ and $\mathbf{B}$, we denote $\langle \mathbf{A}, \mathbf{B} \rangle = \sum_{i,j} \mathbf{A}_{ij} \mathbf{B}_{ij}$ and $\mathbf{A} \otimes \mathbf{B}$ as their Kronecker product.

---

[2]We define the gradient complexity as the number of required stochastic gradient evaluations to achieve a target excess risk, which is closely related to the total computation time.

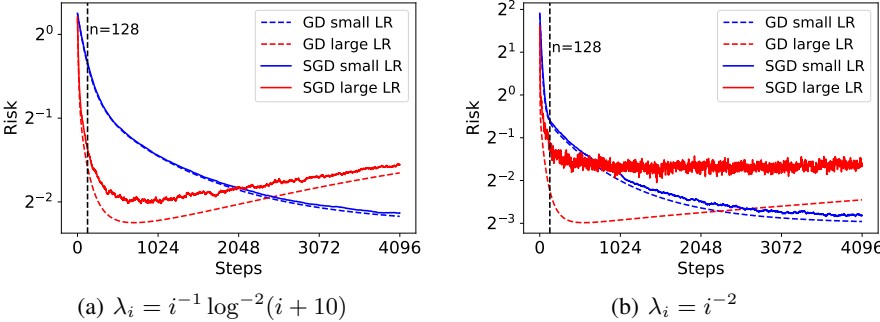

(a) $\lambda_i = i^{-1}\log^{-2}(i+10)$          (b) $\lambda_i = i^{-2}$

Figure 1: Excess risk comparison between SGD and GD with large and small stepsizes. The true parameter $\mathbf{w}^*$ is randomly drawn from $\mathcal{N}(0, \mathbf{I})$ and the model noise variance $\sigma^2 = 1$. The problem dimension is $d = 256$, and we randomly draw $n = 128$ training data. We consider two data covariance with eigenspectrum $\lambda_i = i^{-1}\log^{-2}(i+10)$ and $\lambda_i = i^{-2}$. For SGD, the reported risk is averaged over 100 repeats of the algorithm's randomness. The large stepsize is $\eta = 0.2$ and the small stepsize is $\eta = 0.02$.

## 2 Related Work

**Optimization.** Regarding optimization efficiency, the benefit of SGD is well understood [7, 8, 23, 5, 34, 35, 26]. For example, for strongly convex losses (can be relaxed with certain growth conditions), GD has less iteration complexity, but SGD enjoys less gradient complexity [7, 8]. More recently, it is shown that SGD can converge at an exponential rate in the interpolating regime [26, 23, 5, 34, 35], therefore SGD can match the iteration complexity of GD. Nevertheless, all the above results are regrading the optimization performance; our focus in this paper is to study the generalization performance of SGD (and GD).

**Risk Bounds for Multi-Pass SGD.** The risk bounds of multi-pass SGD are also studied from the operator perspective [30, 22, 28, 25]. The work by Rosasco and Villa [30] focused on *cyclic SGD*, i.e., SGD with multiple passes but fixed sequence on the training data. Their results are limited to small stepsizes ($\gamma = \mathcal{O}(1/n)$), while ours allow constant stepsize. Similar to Lin and Rosasco [22], Pillaud-Vivien et al. [28], Mücke et al. [25], we decompose the population risk of SGD iterates into a risk term caused by batch GD iterates and a fluctuation error term between SGD and GD iterates. But our methods of bounding the fluctuation error are different (see more in Section 5). Moreover, our results are based on different assumptions: Lin and Rosasco [22], Pillaud-Vivien et al. [28], Mücke et al. [25] assumed finiteness on the optimal parameter, and their results only apply to data covariance with a specific type of spectrum (nearly polynomially decaying ones); in contrast, our results assume a Gaussian prior on the optimal parameter (which might not admit a finite norm), and our results are stated as a function of the entire eigenspectrum of the data covariance, thus cover more general data covariance (including those with polynomially decaying spectrum). Lei et al. [21] studied risk bounds for multi-pass SGD with general convex loss. When applied to least square problems, their bounds are cruder than ours.

**Uniform Stability.** Another approach for characterizing the generalization of multi-pass SGD is through *uniform stability* [16, 9, 20, 41, 6]. There are mainly two differences between this and our approach. First, we directly bound the excess risk of SGD; but the uniform stability can only bound the generalization error, there needs an additional triangle inequality to relate excess risk with generalization error plus optimization error (plus approximation error) — this inequality can easily be loose (consider the algorithmic regularization effects). Secondly, the uniform stability bound is also crude. For example, in the non-strongly convex setting, the uniform stability bound for SGD/GD linearly scales with the total optimization length (i.e., sum of stepsizes), which grows as $t$ [16, 9, 20, 41, 6] (this is minimaxly unavoidable according to Zhang et al. [41], Bassily et al. [6]). Notably, Bassily et al. [6] extended the uniform stability approach to the non-convex and smooth setting. We left such an extension of our method as a future work.

## 3 Problem Setup

Let $\mathbf{x}$ be a feature vector in a Hilbert space $\mathcal{H}$ (its dimension is denoted by $d$, which is possibly infinite) and $y \in \mathbb{R}$ be its response, and assume that they jointly follow an unknown population

distribution $\mathcal{D}$. In linear regression problems, the *population risk* of a parameter $\mathbf{w}$ is defined by

$$L_\mathcal{D}(\mathbf{w}) := \frac{1}{2}\mathbb{E}_{(\mathbf{x},y)\sim D}(\langle \mathbf{x}, \mathbf{w}\rangle - y)^2,$$

and the *excess risk* is defined by

$$\mathcal{E}(\mathbf{w}) := L_\mathcal{D}(\mathbf{w}) - \min_\mathbf{w} L_\mathcal{D}(\mathbf{w}) = \frac{1}{2}\|\mathbf{w} - \mathbf{w}^*\|_\mathbf{H}^2, \quad \text{where } \mathbf{H} := \mathbb{E}_\mathcal{D}[\mathbf{x}\mathbf{x}^\top], \qquad (3.1)$$

where $\mathbf{w}^* = \arg\min_\mathbf{w} L_\mathcal{D}(\mathbf{w})$ denotes the global minimizer of the population risk. Additionally, following Zou et al. [42, 43], we assume that the data covariance matrix $\mathbf{H}$ is positive definite. In the statistical learning setting, the population distribution $\mathcal{D}$ is unknown, and one is provided with a set of $n$ training samples, $\mathcal{S} = (\mathbf{x}_i, y_i)_{i=1}^n$, that are drawn independently at random from the population distribution. We also use $\mathbf{X} := (\mathbf{x}_1, \ldots, \mathbf{x}_n)^\top$ and $\mathbf{y} := (y_1, \ldots, y_n)^\top$ to denote the concatenated features and labels, respectively. The linear regression problems aim to find a parameter based on the training set $\mathcal{S}$ that affords a small excess risk.

**Multi-Pass SGD.** We are interested in solving the linear regression problem using multi-pass *stochastic gradient descent* (SGD)[3] with a constant learning rate. The algorithm generates a sequence of iterates $(\mathbf{w}_t)_{t\geq 1}$ according to the following update rule: the initial iterate is $\mathbf{w}_0 = \mathbf{0}$ (which can be assumed without loss of generality); then at each iteration, an example $(\mathbf{x}_{i_t}, y_{i_t})$ is drawn from $\mathcal{S}$ uniformly at random, and the iterate is updated by

$$\mathbf{w}_{t+1} = \mathbf{w}_t - \eta \cdot \mathbf{x}_{i_t}(\mathbf{x}_{i_t}^\top \mathbf{w}_t - y_{i_t}),$$

where $\eta > 0$ is a constant stepsize (i.e., learning rate).

**GD.** Another popular algorithm is *gradient descent* (GD). For the clarity of notations, we use $(\widehat{\mathbf{w}}_t)_{t\geq 1}$ to denote the GD iterates, which follow the following updates:

$$\widehat{\mathbf{w}}_{t+1} = \widehat{\mathbf{w}}_t - \eta \cdot \frac{1}{n}\sum_{i=1}^n \mathbf{x}_i(\mathbf{x}_i^\top \widehat{\mathbf{w}}_t - y_i), \quad \widehat{\mathbf{w}}_0 = \mathbf{0},$$

where $\eta > 0$ is a constant stepsize.

**Definitions and Assumptions.** The eigenvalues of the population data covariance $\mathbf{H}$ is denoted by $(\lambda_i)_{i\geq 1}$, sorted in non-increasing order. Given the training data $(\mathbf{X}, \mathbf{y})$, we define $\boldsymbol{\epsilon} = \mathbf{y} - \mathbf{X}\mathbf{w}^*$ the collection of model noise, $\mathbf{A} = \mathbf{X}\mathbf{X}^\top$ as the Gram matrix, and $\boldsymbol{\Sigma} = n^{-1}\mathbf{X}^\top\mathbf{X}$ as the empirical covariance. Then the *minimum-norm solution* is defined by

$$\widehat{\mathbf{w}} := (\mathbf{X}^\top\mathbf{X})^\dagger\mathbf{X}^\top\mathbf{y} = \mathbf{X}^\top\mathbf{A}^{-1}\mathbf{y}.$$

It is clear that in the interpolation regime, with appropriate stepsizes, both SGD and GD algorithms converge to $\widehat{\mathbf{w}}$ [14, 4].

The assumptions required by our theorems are summarized in below.

**Assumption 3.1** *For the linear regression problem:*

A *The components of $\mathbf{H}^{-1/2}\mathbf{x}$ are independent and $1$-subGaussian.*
B *The response $y$ is generated by $y := \langle \mathbf{w}^*, \mathbf{x}\rangle + \xi$, where $\mathbf{w}^*$ is the ground truth weight vector and $\xi$ is a noise independent of $\mathbf{x}$. Furthermore, the additive noise satisfies $\mathbb{E}[\xi] = 0$, $\mathbb{E}[\xi^2] \leq \sigma^2$.*
C *The ground truth $\mathbf{w}^*$ follows a Gaussian prior $\mathcal{N}(\mathbf{0}, \omega^2 \cdot \mathbf{I})$, where $\omega^2$ is a constant.*
D *The minimum-norm solution $\widehat{\mathbf{w}}$ linearly interpolates all training data, i.e., $y_i = \widehat{\mathbf{w}}^\top\mathbf{x}_i$ for $i \in [n]$.*

Assumptions 3.1A and B are standard for analyzing overparameterized linear regression problem in the benign overfitting regime [4, 33]. Note that Assumption 3.1A is widely made in the analysis of high-dimensional least squares estimations [11, 37, 17]. However, Assumption 3.1A is not standard for analyzing SGD [3, 22, 28, 18, 42]. We conjecture Assumption 3.1A can be relaxed and leave this as a future work. Moreover, Assumption 3.1C is also widely adopted in analyzing least square problems (see, e.g., Ali et al. [2], Dobriban et al. [11], Xu and Hsu [39]). There are also many different conditions being made on the ground truth $\mathbf{w}^*$ in existing works [28] to study the generalization

---

[3]We focus on *SGD with replacement* in this paper. Extending our results to SGD without replacement is an important yet challenging future direction (see more discussions in Section 6).

of SGD (e.g., $\|\mathbf{H}^{1/2-r}\mathbf{w}^*\|_2 \to \infty$ for $r \geq 0$). However, they are not directly comparable to Assumption 3.1C. Finally, Assumption 3.1D holds almost surely when $d > n$, i.e., the number of parameter exceeds the number of data.

In the following, the presented risk bounds will hold (i) with high-probability with respect to the randomness of sampling feature vectors $\mathbf{X}$, and (ii) in expectation with respect to the randomness of multi-pass SGD algorithm, the randomness of sampling additive noise $\boldsymbol{\epsilon}$ and the randomness of the true parameter $\mathbf{w}^*$ as a prior. For these purpose, we will use $\mathbb{E}_{i_t}, \mathbb{E}_{\mathrm{SGD}}, \mathbb{E}_{\mathbf{w}^*}$ to refer to taking expectation with respect to the randomness of sampling data (from the training set) at the $t$-th iteration, the randomness of the entire SGD algorithm (i.e., sampling data at each iteration, $i_1, \ldots, i_t, \ldots$) and the prior distribution of $\mathbf{w}^*$, respectively.

# 4  Main Results

Our first theorem shows that, under the same stepsize and number of iterates, SGD always generalizes worse than GD.

**Theorem 4.1 (Risk decomposition)** *Suppose that Assumption 3.1D holds. Then the excess risk of SGD can be decomposed by*

$$\mathbb{E}_{\mathrm{SGD}}\big[\mathcal{E}(\mathbf{w}_t)\big] = \mathcal{E}(\widehat{\mathbf{w}}_t) + \mathrm{FluctuationError}(\mathbf{w}_t).$$

*Moreover, the fluctuation error is always non-negative.*

**A Risk Comparison.** Theorem 4.1 shows that, in the interpolation regime, SGD affords a strictly larger excess risk than GD, given the same hyperparameters (stepsize $\eta$ and number of iterates $t$). Therefore, despite of a possibly higher computational cost, the optimally tuned GD *dominates* the optimally tuned SGD in terms of the generalization performance. This observation is verified empirically by experiments in Figure 1.

Theorem 4.1 relates the risk of SGD iterates to that of GD iterates. This idea has appeared in earlier literature [22, 28, 25]. However, their decomposition is obtained via Young's inequality (see, e.g., Eq. (13) in Appendix A of Mücke et al. [25]), and is therefore stated as an upper bound on the SGD risk.

Our next theorem is to characterize the fluctuation error of SGD (with respect to GD).

**Theorem 4.2 (Fluctuation error bound)** *Suppose that Assumptions 3.1A, B and D all hold. Then for every $n \geq 1$, $t \geq 1$ and $\eta \leq c/\operatorname{tr}(\mathbf{H})$ for some absolute constant $c$, with probability at least $1 - 1/\operatorname{poly}(n)$, it holds that*

$\mathrm{FluctuationError}(\mathbf{w}_t) \lesssim$

$$\left[ \log(t) \cdot \left( \frac{\operatorname{tr}(\mathbf{H})\log(n)}{t} + \frac{k^\dagger \log^{5/2}(n)}{n^{1/2}t} \right) + \frac{\log^{5/2}(n)\eta}{n^{1/2}} \cdot \sum_{i>k^\dagger} \lambda_i \right] \cdot \min\big\{ \|\widehat{\mathbf{w}}\|_2^2,\ t\eta \cdot \|\widehat{\mathbf{w}}\|_{\boldsymbol{\Sigma}}^2 \big\},$$

*where $k^\dagger \geq 0$ is an arbitrary index (can be infinity).*

We first explain the factor $\min\big\{\|\widehat{\mathbf{w}}\|_2^2,\ t\eta \cdot \|\widehat{\mathbf{w}}\|_{\boldsymbol{\Sigma}}^2\big\}$ in our bound. First of all, when the interpolator $\widehat{\mathbf{w}}$ has a small $\ell_2$-norm, the quantity is automatically small. Furthermore, $\|\widehat{\mathbf{w}}\|_{\boldsymbol{\Sigma}}^2 \lesssim \omega^2 \lesssim 1$ easily holds under mild assumptions on $\mathbf{w}^*$, e.g., Assumption 3.1C. Then, for finite $t$ one can bound the factor with $\min\big\{\|\widehat{\mathbf{w}}\|_2^2,\ t\eta \cdot \|\widehat{\mathbf{w}}\|_{\boldsymbol{\Sigma}}^2\big\} \lesssim \omega^2\eta t$. More interestingly, for SGD with constant stepsize and infinite optimization steps ($t \to \infty$), our risk bound can still vanish, while all risk bounds in prior works [22, 28, 25] are vacuous. To see this, one can consider a sequence of $k^\dagger$, e.g., $k^\dagger = \sqrt{t}$, then it is clear that $\log(t)k^\dagger(t)/t, \sum_{i>k^\dagger(t)} \lambda_i \to 0$ when $t \to \infty$, so the fluctuation error vanishes.

To complement the above results, we provide the following risk bound for GD. We emphasize that any risk bound for GD can be plugged into Theorems 4.1 and 4.2 to obtain a risk bound for SGD.

**Theorem 4.3 (GD risk)** *Suppose that Assumptions 3.1A, B and C all hold. Then for every $n \geq 1$, $t \geq 1$ and $\eta < 1/\|\mathbf{H}\|_2$, with probability at least $1 - 1/\operatorname{poly}(n)$, it holds that*

$$\mathbb{E}_{\mathbf{w}^*,\boldsymbol{\epsilon}}[\mathcal{E}(\widehat{\mathbf{w}}_t)] \lesssim \omega^2 \cdot \left( \frac{\widetilde{\lambda}^2}{n^2} \cdot \sum_{i \leq k^*} \frac{1}{\lambda_i} + \sum_{i > k^*} \lambda_i \right) + \sigma^2 \cdot \left( \frac{k^*}{n} + \frac{n}{\widetilde{\lambda}^2} \sum_{i>k^*} \lambda_i^2 \right),$$

*where $k^* := \min\{k : n\lambda_{k+1} \leq \frac{n}{\eta t} + \sum_{i>k} \lambda_i\}$ and $\widetilde{\lambda} := \frac{n}{\eta t} + \sum_{i>k^*} \lambda_i$.*

The bound presented in Theorem 4.3 is comparable to that for ridge regression established by Tsigler and Bartlett [33] and will be much better than the bound of single-pass SGD when the signal-to-noise ratio is large [43, Theorem 5.1], e.g., $\omega^2 \gg \sigma^2$. In fact, Theorem 4.3 is proved via a reduction to ridge regression results. In particular, the quantity $n/(\eta t)$ for GD is an analogy to the regularization parameter $\lambda$ for ridge regression [40, 29, 36, 2]. As a final remark, the assumption that $\mathbf{w}^*$ follows a Gaussian prior is the main concealing in Theorem 4.3 (which is not required by Tsigler and Bartlett [33]). The Gaussian prior on $\mathbf{w}^*$ is known to allow a connection between early stopped GD with ridge regression [2]. We conjecture that this assumption is not necessary and potentially removable.

Combining Theorems 4.1, 4.2 and 4.3, we obtain the following risk bound for multi-pass SGD:

**Corollary 4.4** *Suppose that Assumptions 3.1A, B, C and D all hold. Then with probability at least $1 - 1/\mathrm{poly}(n)$, it holds that*

$$\mathbb{E}_{\mathrm{SGD},\mathbf{w}^*,\boldsymbol{\epsilon}}\left[\mathcal{E}(\mathbf{w}_t)\right] \lesssim \omega^2 \cdot \left(\frac{\widetilde{\lambda}^2}{n^2} \cdot \sum_{i \leq k^*} \frac{1}{\lambda_i} + \sum_{i>k^*} \lambda_i\right) + \sigma^2 \cdot \left(\frac{k^*}{n} + \frac{n}{\widetilde{\lambda}^2} \sum_{i>k^*} \lambda_i^2\right)$$

$$+ \eta \cdot \left[\log(t) \cdot \left(\mathrm{tr}(\mathbf{H})\log(n) + \frac{k^\dagger \log^{5/2}(n)}{n^{1/2}}\right) + \frac{\log^{5/2}(n)t\eta}{n^{1/2}} \cdot \sum_{i>k^\dagger} \lambda_i\right)\right]$$

$$\cdot \min\left\{(t\eta)^{-1} \cdot \left(n\omega^2 + \sigma^2 \mathrm{tr}(\mathbf{A}^{-1})\right), \omega^2 \mathrm{tr}(\mathbf{H}) + \sigma^2\right\},$$

*where $k^\dagger$ is an arbitrary index, $k^* := \min\{k : n\lambda_{k+1} \leq \frac{n}{\eta t} + \sum_{i>k} \lambda_i\}$ and $\widetilde{\lambda} := \frac{n}{\eta t} + \sum_{i>k^*} \lambda_i$.*

**Comparison with Existing Results.** We now discuss relationships between our bound and existing ones for multi-pass SGD [22, 28, 25]. First, we highlight that our bound is *problem-dependent* in the sense that the bound is stated as a function of the spectrum of data covariance; in contrast, existing papers only provide a minimax analysis for multi-pass SGD. Secondly, we rely on a different set of assumptions from the aforementioned papers. In particular, Pillaud-Vivien et al. [28] requires a *source condition* on the data covariance (e.g. $\|\mathbf{H}^{1/2-r}\mathbf{w}^*\|_2 < \infty$ for some constant $r \geq 0$), and Lin and Rosasco [22], Mücke et al. [25] require an *effective dimension* to be small, but our results are more general regarding the data covariance. Moreover, we assume $\mathbf{w}^*$ follows a Gaussian prior (Assumption 3.1C), which is also not directly comparable to the *source condition* in existing works.

Bartlett et al. [4] showed that OLS generalizes in the so called *benign overfitting* regime. Since when $t \to \infty$, SGD (with constant stepsize) converges to OLS, it would be interesting to compare the SGD solution with OLS in such a regime. We will do so with the following Corollary 4.5.

**Corollary 4.5** *Suppose that Assumptions 3.1A, B, C and D all hold. Assume the spectrum of $\mathbf{H}$ satisfies $\lambda_i = i^{-1}\log(i+1)^{-\beta}$ for some absolute constant $\beta > 1$, then with probability at least $1 - 1/\mathrm{poly}(n)$, there exists a choice of $t$ and $\eta$ such that*

$$\mathbb{E}_{\mathrm{SGD},\mathbf{w}^*,\boldsymbol{\epsilon}}[\mathcal{E}(\mathbf{w}_t)] \lesssim \omega^2 \cdot \log(n)^{1-\beta} + \sigma^2 \log(n)^{-\beta}.$$

*Besides, for any fixed stepsize $\eta$, we have*

$$\lim_{t\to\infty} \mathbb{E}_{\mathrm{SGD},\mathbf{w}^*,\boldsymbol{\epsilon}}[\mathcal{E}(\mathbf{w}_t)] \lesssim \omega^2 \cdot \log(n)^{1-\beta} + \sigma^2 \log(n)^{-1}.$$

As a sanity check, our bound for $t \to \infty$ matches the upper and lower bounds on the excess risk of OLS (which can be obtained by setting $\lambda = 0$ in Theorem 1 and Lemmas 2 & 3 in Tsigler and Bartlett [33]). Moreover, Corollary 4.5 suggests that the excess risk achieved by multipass SGD is *always no worse than* that of OLS, and could be *strictly smaller than* that of OLS when $\beta > 0$. This demonstrates the benefit of multi-pass SGD over OLS.

The following corollary characterizes the risk of multi-pass SGD for data covariance with a polynomially decaying spectrum.

**Corollary 4.6** *Suppose that Assumptions 3.1A, B, C and D all hold. Assume the spectrum of $\mathbf{H}$ decays polynomially, i.e., $\lambda_i = i^{-1-r}$ for some absolute constant $r > 0$, then with probability at least $1 - 1/\mathrm{poly}(n)$, it holds that*

$$\mathbb{E}_{\mathbf{w}^*,\boldsymbol{\epsilon}}[\mathcal{E}(\widehat{\mathbf{w}}_t)] \lesssim \omega^2 \cdot (t\eta)^{-r/(r+1)} + \sigma^2 \cdot \frac{(t\eta)^{1/(r+1)}}{n},$$

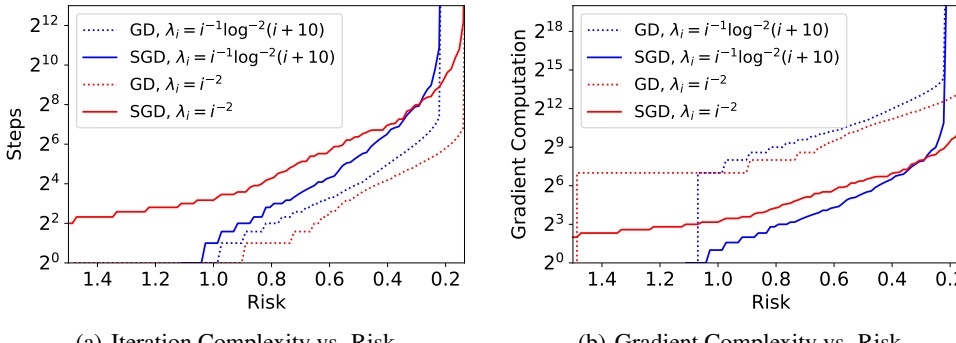

(a) Iteration Complexity vs. Risk         (b) Gradient Complexity vs. Risk

Figure 2: Iteration and gradient complexity comparison between SGD and GD. The curves report the minimum number of steps/gradients for each algorithm (with an optimally tuned stepsize) to achieve a targeted risk. Experiment setup is the same as that in Figure 1.

$$\mathbb{E}_{\mathrm{SGD},\mathbf{w}^*,\boldsymbol{\epsilon}}[\mathcal{E}(\mathbf{w}_t)] \lesssim \omega^2 \cdot (t\eta)^{-r/(r+1)} + \sigma^2 \cdot \frac{(t\eta)^{1/(r+1)}}{n}$$
$$+ (\omega^2 + \sigma^2) \cdot \eta \cdot \log(t) \cdot \left[\log(n) + \frac{\log^{5/2}(n)}{n^{1/2}} \cdot (t\eta)^{1/(r+1)}\right].$$

Corollary 4.6 provides concrete excess risk bounds for SGD and GD, based on which we can make a comparison between SGD and GD in terms of their iteration and gradient complexities. For simplicity, in the following discussion, we assume that $\omega^2 \eqsim \sigma^2 \eqsim 1$. Then choosing $t\eta \eqsim n$ minimizes the upper bound for GD risk and yields the $O(n^{-r/(r+1)})$ rate. Here GD can employ a constant stepsize. Similarly, SGD can match the GD's rate, $O(n^{-r/(r+1)})$, by setting $t\eta \eqsim n$ and

$$\eta \lesssim \log^{-1}(t) \cdot \min\{\log^{-1}(n) \cdot n^{-\frac{r}{r+1}}, \ \log^{-\frac{5}{2}}(n) \cdot n^{-\frac{1}{2}}\}. \tag{4.1}$$

The above implies that SGD (fixed stepsize, last iterate) can only cooperate with small stepsize.

**Iteration Complexity.** We first compare GD and SGD in terms of the iteration complexity. To reach the optimal rate, GD can employ a constant stepsize and set the number of iterates to be $t \eqsim n$. However, in order to shelve the fluctuation error, the stepsize of SGD cannot be large, as required by (4.1). More precisely, in order to match the optimal rate, SGD needs to use a small stepsize, $\eta \eqsim n/t$, with a large number of iterates,

$$t \eqsim \begin{cases} \log(n) \cdot n^{1+\frac{r}{r+1}} = \widetilde{\mathcal{O}}(n^{1+\frac{r}{r+1}}), & r > 1; \\ \log^{3.5}(n) \cdot n^{1.5} = \widetilde{\mathcal{O}}(n^{1.5}), & r \le 1. \end{cases}$$

It can be seen that the iteration complexity of SGD is much worse than that of GD. This result is empirically verified by Figure 2 (a).

**Gradient Complexity.** We next compare GD and SGD in terms of the gradient complexity. Recall that for each iterate, GD computes $n$ gradients but SGD only computes 1 gradient. Therefore, to reach the optimal rate, the total number of gradient computed by GD needs to be $\Theta(n^2)$, but that computed by SGD is only $\widetilde{\mathcal{O}}(n^{\max\{(2r+1)/(r+1),1.5\}})$. Thus, the gradient complexity of SGD is better than that of GD by a factor of $\widetilde{\mathcal{O}}(n^{\min\{0.5,1/(r+1)\}})$. This result is empirically verified by Figure 2 (b).

## 5 Overview of the Proof Technique

In this section, we will provide an overview of our proof technique and sketch the proof of Theorems 4.1 and 4.2. The remaining proof is deferred to Appendix.

Our proof technique is inspired by the operator methods for analyzing single-pass SGD [3, 10, 18, 19, 27, 13, 42, 38]. In particular, they track an error *matrix*, $(\mathbf{w}_t - \mathbf{w}^*) \otimes (\mathbf{w}_t - \mathbf{w}^*)$ that keeps richer information than the error norm $\|\mathbf{w}_t - \mathbf{w}^*\|_2^2$. For single-pass SGD where each data is used only once, the resulted iterates enjoy a simple dependence on history that allows an easy calculation of the expected error matrix (with respect to the randomness of data generation). However for

multi-pass SGD, a data might be used multiple times, which prevents us from tracking the expected error matrix directly. Instead, a trackable analogy to the error matrix is the *empirical error matrix*, $(\mathbf{w}_t - \widehat{\mathbf{w}}) \otimes (\mathbf{w}_t - \widehat{\mathbf{w}})$ where $\widehat{\mathbf{w}}$ is the minimum norm interpolator. More precisely, note that

$$\mathbf{w}_{t+1} - \widehat{\mathbf{w}} = \mathbf{w}_t - \widehat{\mathbf{w}} - \eta \cdot (\mathbf{x}_{i_t}\mathbf{x}_{i_t}^\top \mathbf{w}_t - \mathbf{x}_{i_t}\mathbf{x}_{i_t}^\top \widehat{\mathbf{w}}) = (\mathbf{I} - \eta\mathbf{x}_{i_t}\mathbf{x}_{i_t}^\top)(\mathbf{w}_t - \widehat{\mathbf{w}}). \quad (5.1)$$

Therefore the expected (over the algorithm's randomness) empirical error matrix updates as follows:

$$\text{let} \quad \mathbf{E}_t := \mathbb{E}_{\text{SGD}}\big[(\mathbf{w}_t - \widehat{\mathbf{w}})(\mathbf{w}_t - \widehat{\mathbf{w}})^\top\big], \text{ then } \mathbf{E}_{t+1} = \mathbb{E}_{i_t}\big[(\mathbf{I} - \eta\mathbf{x}_{i_t}\mathbf{x}_{i_t}^\top)\mathbf{E}_t(\mathbf{I} - \eta\mathbf{x}_{i_t}\mathbf{x}_{i_t}^\top)\big]$$

Let $\boldsymbol{\Sigma} := \frac{1}{n}\mathbf{X}^\top\mathbf{X}$ be the empirical covariance matrix. We then follow the operator method [42] to define the following operators on symmetric matrices (e.g., $\mathbf{J}$):

$$\mathcal{G} \circ \mathbf{J} := (\mathbf{I} - \eta\boldsymbol{\Sigma})\mathbf{J}(\mathbf{I} - \eta\boldsymbol{\Sigma}), \quad \mathcal{M} \circ \mathbf{J} := \mathbb{E}_{\text{SGD}}[\mathbf{x}_{i_t}\mathbf{x}_{i_t}^\top \mathbf{J}\mathbf{x}_{i_t}\mathbf{x}_{i_t}^\top], \quad \widetilde{\mathcal{M}} \circ \mathbf{J} := \boldsymbol{\Sigma}\mathbf{J}\boldsymbol{\Sigma}.$$

Based on these operators, we can obtain a close form update rule for $\mathbf{E}_t$:

$$\mathbf{E}_t = \mathcal{G} \circ \mathbf{E}_{t-1} + \eta^2 \cdot (\mathcal{M} - \widetilde{\mathcal{M}}) \circ \mathbf{E}_{t-1} = \underbrace{\mathcal{G}^t \circ \mathbf{E}_0}_{\boldsymbol{\Theta}_1} + \underbrace{\eta^2 \cdot \sum_{k=0}^{t-1} \mathcal{G}^{t-1-k} \circ (\mathcal{M} - \widetilde{\mathcal{M}}) \circ \mathbf{E}_k}_{\boldsymbol{\Theta}_2}. \quad (5.2)$$

Here the first term $\boldsymbol{\Theta}_1 := (\mathbf{I} - \eta\boldsymbol{\Sigma})^t \mathbf{E}_0 (\mathbf{I} - \eta\boldsymbol{\Sigma})^t = (\widehat{\mathbf{w}}_t - \widehat{\mathbf{w}})(\widehat{\mathbf{w}}_t - \widehat{\mathbf{w}})^\top$ is exactly the error matrix caused by GD iterates (with stepsize $\eta$ and iteration number $t$), and the second term $\boldsymbol{\Theta}_2$ is a *fluctuation matrix* that captures the deviation of $\mathbf{w}_t$ with respect to a corresponding GD iterate $\widehat{\mathbf{w}}_t$. We remark that the expected error matrix $\mathbf{E}_t$ contains all information of $\mathbf{w}_t$.

**Risk Decomposition (Theorem 4.1).** The following fact is clear from the update rule (5.1).

**Fact 5.1** *The GD iterates satisfy* $\widehat{\mathbf{w}}_{t+1} - \widehat{\mathbf{w}} = (\mathbf{I} - \eta\boldsymbol{\Sigma})(\widehat{\mathbf{w}}_t - \widehat{\mathbf{w}})$ *and* $\mathbb{E}_{\text{SGD}}[\mathbf{w}_t - \widehat{\mathbf{w}}] = \widehat{\mathbf{w}}_t - \widehat{\mathbf{w}}$.

Based on Fact 5.1 and (5.2), we have

$$\mathbb{E}_{\text{SGD}}[(\mathbf{w}_t - \mathbf{w}^*)(\mathbf{w}_t - \mathbf{w}^*)^\top]$$
$$= \mathbf{E}_t + (\widehat{\mathbf{w}} - \mathbf{w}^*)(\widehat{\mathbf{w}}_t - \widehat{\mathbf{w}})^\top + (\widehat{\mathbf{w}}_t - \widehat{\mathbf{w}})(\widehat{\mathbf{w}} - \mathbf{w}^*)^\top + (\widehat{\mathbf{w}} - \mathbf{w}^*)(\widehat{\mathbf{w}} - \mathbf{w}^*)^\top$$
$$= \boldsymbol{\Theta}_1 + (\widehat{\mathbf{w}} - \mathbf{w}^*)(\widehat{\mathbf{w}}_t - \widehat{\mathbf{w}})^\top + (\widehat{\mathbf{w}}_t - \widehat{\mathbf{w}})(\widehat{\mathbf{w}} - \mathbf{w}^*)^\top + (\widehat{\mathbf{w}} - \mathbf{w}^*)(\widehat{\mathbf{w}} - \mathbf{w}^*)^\top + \boldsymbol{\Theta}_2$$
$$= (\widehat{\mathbf{w}}_t - \mathbf{w}^*)(\widehat{\mathbf{w}}_t - \mathbf{w}^*)^\top + \boldsymbol{\Theta}_2,$$

where $\boldsymbol{\Theta}_1$ and $\boldsymbol{\Theta}_2$ are defined in (5.2) and the last equality is due to $\boldsymbol{\Theta}_1 = (\widehat{\mathbf{w}}_t - \widehat{\mathbf{w}})(\widehat{\mathbf{w}}_t - \widehat{\mathbf{w}})^\top$. Also note that

$$\mathbb{E}_{\text{SGD}}[\mathcal{E}(\mathbf{w}_t)] = \frac{1}{2}\mathbb{E}_{\text{SGD}}\big[\|\mathbf{w}_t - \mathbf{w}^*\|_\mathbf{H}^2\big] = \frac{1}{2}\big\langle \mathbb{E}_{\text{SGD}}[(\mathbf{w}_t - \mathbf{w}^*)(\mathbf{w}_t - \mathbf{w}^*)^\top], \mathbf{H}\big\rangle.$$

Combining these two inequalities proves Theorem 4.1:

$$\mathbb{E}_{\text{SGD}}[\mathcal{E}(\mathbf{w}_t)] = \underbrace{\frac{1}{2}\|\widehat{\mathbf{w}}_t - \mathbf{w}^*\|_\mathbf{H}^2}_{\text{GD error}} + \underbrace{\frac{\eta^2}{2} \cdot \sum_{k=0}^{t-1} \big\langle \mathcal{G}^{t-1-k} \circ (\mathcal{M} - \widetilde{\mathcal{M}}) \circ \mathbf{E}_k, \mathbf{H}\big\rangle}_{\text{Fluctuation error}}. \quad (5.3)$$

Finally, the fluctuation error is non-negative because both $\mathcal{G}$ and $\mathcal{M} - \widetilde{\mathcal{M}}$ are PSD mappings.

**Bounding the Fluctuation Error (Theorem 4.2).** There are several challenges in the analysis of fluctuation error: (1) it is difficult to characterize the matrix $(\mathcal{M} - \widetilde{\mathcal{M}}) \circ \mathbf{E}_k$ since the matrix $\mathbf{E}_k$ is unknown; (2) the operator $\mathcal{G}$ involves an exponential decaying term with respect to the empirical covariance matrix $\boldsymbol{\Sigma}$, which does not commute with the population covariance matrix $\mathbf{H}$.

To address the first problem, we note that the operators $\widetilde{\mathcal{M}}, \mathcal{G}, \mathcal{M}$ are PSD mappings and enjoy commutative property and then obtain the following result:

$$\text{FluctuationError} \leq \frac{\eta^2}{2} \cdot \sum_{k=0}^{t-1} \langle \mathcal{M} \circ \mathcal{G}^{t-1-k} \circ \mathbf{H}, \mathbf{E}_k\rangle. \quad (5.4)$$

Now, the input of the operator $\mathcal{M} \circ \mathcal{G}^{t-1-k}$ will not be an unknown matrix but a fixed one (i.e., $\mathbf{H}$), and the remaining effort will be focusing on characterizing $\mathcal{M} \circ \mathcal{G}^k \circ \mathbf{H}$. Applying the definitions of $\mathcal{M}$ and $\mathcal{G}$ implies

$$\mathcal{M} \circ \mathcal{G}^k \circ \mathbf{H} = \mathbb{E}_i\big[\mathbf{x}_i\mathbf{x}_i^\top (\mathbf{I} - \eta\boldsymbol{\Sigma})^k \mathbf{H}(\mathbf{I} - \eta\boldsymbol{\Sigma})^k \mathbf{x}_i\mathbf{x}_i^\top\big]. \quad (5.5)$$

Then our idea is to first prove an uniform upper bound on the quantity $\mathbf{x}_i^\top (\mathbf{I} - \eta\mathbf{\Sigma})^k \mathbf{H}(\mathbf{I} - \eta\mathbf{\Sigma})^k \mathbf{x}_i$ for all $i \in [n]$ (e.g., denoted as $U(k, \eta, n)$), then it can be naturally obtained that

$$\mathcal{M} \circ \mathcal{G}^k \circ \mathbf{H} \preceq U(k, \eta, n) \cdot \mathbb{E}_i[\mathbf{x}_i\mathbf{x}_i^\top] = U(k, \eta, n) \cdot \mathbf{\Sigma}, \tag{5.6}$$

then we will only need to characterize the inner product $\langle \mathbf{E}_k, \mathbf{\Sigma} \rangle$ in (5.4), which can be understood as the optimization error at the $k$-th iteration.

In order to precisely characterize $U(k, \eta, n)$, we encounter the second problem that the population covariance $\mathbf{H}$ and empirical covariance $\mathbf{\Sigma}$ are not commute, thus the exponential decaying term $(\mathbf{I} - \eta\mathbf{\Sigma})^k$ will not be able to fully decrease $\mathbf{H}$ since some components of $\mathbf{H}$ may lie in the small eigenvalue directions of $\mathbf{\Sigma}$. Therefore, we consider the following decomposition

$$\mathbf{x}_i^\top (\mathbf{I} - \eta\mathbf{\Sigma})^k \mathbf{H}(\mathbf{I} - \eta\mathbf{\Sigma})^k \mathbf{x}_i = \underbrace{\mathbf{x}_i^\top (\mathbf{I} - \eta\mathbf{\Sigma})^k \mathbf{\Sigma}(\mathbf{I} - \eta\mathbf{\Sigma})^k \mathbf{x}_i}_{} + \underbrace{\mathbf{x}_i^\top (\mathbf{I} - \eta\mathbf{\Sigma})^k (\mathbf{H} - \mathbf{\Sigma})(\mathbf{I} - \eta\mathbf{\Sigma})^k \mathbf{x}_i}_{}.$$

Then for $\Theta_1$, it can be seen that the decaying term $(\mathbf{I} - \eta\mathbf{\Sigma})^k$ is commute with $\mathbf{\Sigma}$ thus can successfully make it decrease. For $\Theta_2$, we will view the difference $\mathbf{H} - \mathbf{\Sigma}$ as the component of $\mathbf{H}$ that cannot be effectively decreased by $(\mathbf{I} - \eta\mathbf{\Sigma})^k$, which will be small as $n$ increases. More specifically, we can get the following upper bound on $\Theta_1$.

**Lemma 5.2** *If the stepsize satisfies $\gamma \leq c/\operatorname{tr}(\mathbf{H})$ for some small absolute constant $c$, then with probability at least $1 - 1/\operatorname{poly}(n)$, it holds that $\Theta_1 \lesssim \operatorname{tr}(\mathbf{H}) \cdot \log(n) \cdot \min\left\{\frac{1}{(k+1)\eta}, \|\mathbf{H}\|_2\right\}$.*

For $\Theta_2$, we will rewrite $\mathbf{x}_i$ as $\mathbf{e}_i^\top \mathbf{X}$ where $\mathbf{e}_i \in \mathbb{R}^n$ and $\mathbf{X} \in \mathbb{R}^{n \times d}$, then

$$\Theta_2 = \mathbf{e}_i^\top \mathbf{X}(\mathbf{I} - \eta\mathbf{\Sigma})^k (\mathbf{H} - \mathbf{\Sigma})(\mathbf{I} - \eta\mathbf{\Sigma})^k \mathbf{X}^\top \mathbf{e}_i \leq \|\mathbf{e}_i^\top \mathbf{X}(\mathbf{I} - \eta\mathbf{\Sigma})^k\|_2^2 \cdot \|\mathbf{H} - \mathbf{\Sigma}\|_2. \tag{5.7}$$

Then since $\mathbf{X}$ and $\mathbf{\Sigma}$ have the same column eigenspectrum, we can fully unleash the decaying power of the term $(\mathbf{I} - \eta\mathbf{\Sigma})^k$ on $\mathbf{X}$. Further note the that the row space of $\mathbf{X}$ is uniform distributed (corresponding to the index of training data), which is independent of $\mathbf{e}_i$. This implies that we can adopt standard concentration arguments with covering on $n$ fixed vectors $\{\mathbf{e}_i\}_{i=1}^n$ to prove a sharp high probability upper bound (compared to the naive worst-case upper bound). Consequently, we state the upper bound on $\Theta_2$ in the following lemma.

**Lemma 5.3** *For every $i \in [n]$ and $k^* \in [d]$, it holds with probability at least $1 - 1/\operatorname{poly}(n)$ that $\Theta_2 \lesssim \frac{\log^{5/2}(n)}{n^{1/2}} \cdot \left(\frac{k^*}{(k+1)\eta} + \sum_{i > k^*} \lambda_i\right)$.*

## 6 Conclusion and Discussion

In this paper, we establish an instance-dependent excess risk bound of multi-pass SGD for interpolating least square problems. The key takeaways include: (1) the excess risk of SGD is *always* worse than that of GD, given the same setup of stepsize and iteration number; (2) in order to achieve the same level of excess risk, SGD requires more iterations than GD; and (3) however, the gradient complexity of SGD can be better than that of GD. The proposed technique for analyzing multi-pass SGD could be of broader interest. The code and data for our experiments can be found on Github [4].

Several interesting problems are left for future exploration:

**A problem-dependent excess risk lower bound** could be useful to help understand the sharpness of our excess risk upper bound for multi-pass SGD and establish a clear separation between SGD and GD in terms of iteration and gradient complexity. The challenge here is mainly from the fact that the empirical covariance matrix $\mathbf{\Sigma}$ does not commute with the population covariance matrix $\mathbf{H}$. In particular, one needs to develop an even sharper characterization on the quantity $\mathcal{M} \circ \mathcal{G}^k \circ \mathbf{H}$ (see (5.5)); more precisely, a sharp lower bound on $\mathbf{x}_i^\top (\mathbf{I} - \eta\mathbf{\Sigma})^k \mathbf{H}(\mathbf{I} - \eta\mathbf{\Sigma})^k \mathbf{x}_i$ is required.

**SGD with decaying stepsizes** could potentially improve the generalization performance of SGD with a constant stepsize. In this regard, most of our analysis can be extended to SGD/GD with decaying stepsizes. For example, Theorem 4.1 directly holds even for varying stepsizes, and Theorem 4.3 also holds under a small modification, i.e., changing $t\eta$ to $\sum_{k=0}^{t-1} \eta_k$. However, our bounds on the fluctuation error may be more subtle to adapt. We conjecture that our analysis idea can still be applied, but the detailed calculations will depend on the particular stepsize scheduler of interests.

---

[4]https://github.com/uclaml/multipass-SGD

**Multi-pass SGD without replacement** is a more practical SGD variant than the multi-pass SGD with replacement studied in this work. The key difference is that, the former does not pass training data independently (since each data must be used for equal times). In terms of optimization complexity, it has already been demonstrated in theory that multi-pass SGD without replacement (e.g., SGD with single shuffle or random shuffle) outperforms multi-pass SGD with replacement [15, 31, 1]. However, in terms of generalization, whether or not multi-pass SGD without replacement can outperform multi-pass SGD with replacement is still an open problem, as there lacks a sharp excess risk analysis for multi-pass SGD without replacement. The techniques presented in this paper can shed light on this direction.

## Acknowledgments and Disclosure of Funding

We would like to thank the anonymous reviewers and area chairs for their helpful comments. This work was done when DZ was a Ph.D. student at UCLA. DZ is partially supported by Bloomberg data science Ph.D. fellowship and his startup funding in the institute of data science, the University of Hong Kong. JW and VB are supported by the Defense Advanced Research Projects Agency (DARPA) under Contract No. HR00112190130. QG is partially supported by the National Science Foundation award IIS-1906169 and IIS-2008981. SK acknowledges funding from the Office of Naval Research under award N00014-22-1-2377 and the National Science Foundation Grant under award CCF-1703574. The views and conclusions contained in this paper are those of the authors and should not be interpreted as representing any funding agencies.

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
