}$, where $\mathbf{A} := \mathbf{X}\mathbf{X}^\top$ is the gram matrix. Then we can reformulate $\widehat{\mathbf{w}}_t$ by

$$\widehat{\mathbf{w}}_t = \widehat{\mathbf{w}} - (\mathbf{I} - \eta\boldsymbol{\Sigma})^t(\widehat{\mathbf{w}}_0 - \widehat{\mathbf{w}}) = \left(\mathbf{I} - (\mathbf{I} - \eta\boldsymbol{\Sigma})^t\right)\mathbf{X}^\top \mathbf{A}^{-1}\mathbf{y} = \mathbf{X}^\top \left(\mathbf{I} - (\mathbf{I} - \eta n^{-1}\mathbf{A})^t\right)\mathbf{A}^{-1}\mathbf{y}.$$

Denote $\widetilde{\mathbf{A}} := \mathbf{A}\left(\mathbf{I} - (\mathbf{I} - \eta n^{-1}\mathbf{A})^t\right)^{-1}$, the excess risk of $\widehat{\mathbf{w}}_t$ is

$$\mathcal{E}(\widehat{\mathbf{w}}_t) = \frac{1}{2}\left\|\mathbf{X}^\top \widetilde{\mathbf{A}}^{-1}\mathbf{y} - \mathbf{w}^*\right\|_{\mathbf{H}}^2 = \underbrace{\frac{1}{2}\left\|\mathbf{w}^*(\mathbf{I} - \mathbf{X}^\top \widetilde{\mathbf{A}}^{-1}\mathbf{X})\right\|_{\mathbf{H}}^2}_{\text{BiasError}} + \underbrace{\frac{1}{2}\left\|\mathbf{X}^\top \widetilde{\mathbf{A}}^{-1}\boldsymbol{\epsilon}\right\|_{\mathbf{H}}^2}_{\text{VarError}}. \qquad \text{(A.1)}$$

The remaining proof will be relates the excess risk of early stopped GD to that of ridge regression with certain regularization parameters. In particular, note that the excess risk of the ridge regression solution with parameter $\lambda$ is $\frac{1}{2}\|\mathbf{X}^\top(\mathbf{A} + \lambda\mathbf{I})^{-1}\mathbf{y} - \mathbf{w}^*\|_{\mathbf{H}}^2$. Then it remains to show the relationship between $\widetilde{\mathbf{A}}$ and $\mathbf{A} + \lambda\mathbf{I}$, which is illustrated in the following lemma.

**Lemma A.1** *There is a constant $c > 0$ such that for every $\eta \leq c/\lambda_1$ and $t > 0$, it holds that* $\frac{1}{2}\left(\mathbf{A} + \frac{n}{\eta t}\mathbf{I}\right) \preceq \widetilde{\mathbf{A}} \preceq \mathbf{A} + \frac{2n}{t\eta} \cdot \mathbf{I}$.

Then, the lower bound of $\widetilde{\mathbf{A}}$ will be applied to prove the upper bound of variance error of GD, as shown in (A.1), which is at most four times the variance error achieved by the ridge regression with $\lambda = n/(\eta t)$. The upper bound of $\widetilde{\mathbf{A}}$ will be applied to prove the upper bound of the bias error of GD, which is at most the bias error achieved by ridge regression with $\lambda = 2n/(\eta t)$. Finally, we can apply the prior work [33, Theorem 1] on the excess risk analysis for ridge regression to complete the proof for bounding the bias and variance errors separately.

# B  Risk Bound for the Fluctuation Error

We first state the following properties of the operators $\mathcal{G}$, $\mathcal{M}$, and $\widetilde{\mathcal{M}}$, which are essential in the subsequent analysis:

- **PSD mapping:** for every PSD matrix $\mathbf{J}$, $\mathcal{M} \circ \mathbf{J}$, $(\mathcal{M} - \widetilde{\mathcal{M}}) \circ \mathbf{J}$ and $\mathcal{G} \circ \mathbf{J}$ are all PSD matrices.
- **Commutative property:** for two PSD matrices $\mathbf{B}_1$ and $\mathbf{B}_2$, we have

$$\langle \mathcal{G} \circ \mathbf{B}_1, \mathbf{B}_2 \rangle = \langle \mathbf{B}_1, \mathcal{G} \circ \mathbf{B}_2 \rangle, \ \langle \mathcal{M} \circ \mathbf{B}_1, \mathbf{B}_2 \rangle = \langle \mathbf{B}_1, \mathcal{M} \circ \mathbf{B}_2 \rangle, \ \langle \widetilde{\mathcal{M}} \circ \mathbf{B}_1, \mathbf{B}_2 \rangle = \langle \mathbf{B}_1, \widetilde{\mathcal{M}} \circ \mathbf{B}_2 \rangle$$

## B.1  Proof of Inequality (5.4)

**Lemma B.1** *The fluctuation error satisfies*

$$\text{FluctuationError} \leq \frac{\eta^2}{2} \cdot \sum_{k=0}^{t-1} \langle \mathcal{M} \circ \mathcal{G}^{t-1-k} \circ \mathbf{H}, \mathbf{E}_k \rangle.$$

**Proof.** [Proof of Lemma B.1] By Lemma 5.3, we have

$$\text{FluctuationError} = \frac{\eta^2}{2} \cdot \sum_{k=0}^{t-1} \langle \mathcal{G}^{t-1-k} \circ (\mathcal{M} - \widetilde{\mathcal{M}}) \circ \mathbf{E}_k, \mathbf{H} \rangle.$$

Then note that $\mathcal{M}$, $\mathcal{M} - \widetilde{\mathcal{M}}$ and $\mathcal{G}$ are the PSD mapping. Then we have

$$\mathcal{G}^{t-1-k} \circ (\mathcal{M} - \widetilde{\mathcal{M}}) \circ \mathbf{E}_k \preceq \mathcal{G}^{t-1-k} \circ \mathcal{M} \circ \mathbf{E}_k$$

for all $k \geq 0$. Further using the commutative property of $\mathcal{G}$ and $\mathcal{M}$, we have

$$\langle \mathcal{G}^{t-1-k} \circ \mathcal{M} \circ \mathbf{E}_k, \mathbf{H} \rangle = \langle \mathcal{M} \circ \mathcal{G}^{t-1-k} \circ \mathbf{H}, \mathbf{E}_k \rangle.$$

This completes the proof.

∎

## B.2 Proof of Lemma 5.2

We first present the following two useful lemmas.

**Lemma B.2 (Theorem 9 in Bartlett et al. [4])** *There is an absolute constant $c$ such that for any $\delta \in (0, 1)$ with probability at least $1 - \delta$,*

$$\|\mathbf{\Sigma} - \mathbf{H}\|_2 \leq c\|\mathbf{H}\|_2 \cdot \max\left\{\sqrt{\frac{r(\mathbf{H})}{n}}, \frac{r(\mathbf{H})}{n}, \sqrt{\frac{\log(1/\delta)}{n}}, \frac{\log(1/\delta)}{n}\right\},$$

*where $r(\mathbf{H}) = \sum_i \lambda_i/\lambda_1$.*

**Lemma B.3 (Lemma 22 in [4])** *There is a universal constant $c$ such that for any independent, mean zero, $\sigma$-subexponential random variables $\xi_1, \ldots, \xi_n$, any $\mathbf{a} = (a_1, \ldots, a_n)$ and any $t \geq 0$,*

$$\mathbb{P}\left(\left|\sum_{i=1}^{n} a_i\xi_i\right| \geq t\right) \leq 2\exp\left[-c\min\left(\frac{t^2}{\sigma^2\|\mathbf{a}\|_2^2}, \frac{t}{\sigma\|\mathbf{a}\|_\infty}\right)\right].$$

**Proof.** [Proof of Lemma 5.2] Note that $(1 - x)^k \leq 1/[x(k + 1)]$ for all $k > 0$ and $x \in (0, 1)$, we have

$$(\mathbf{I} - \eta\mathbf{\Sigma})^k\mathbf{\Sigma}(\mathbf{I} - \eta\mathbf{\Sigma})^k = \mathbf{\Sigma}(\mathbf{I} - \eta\mathbf{\Sigma})^{2k} \preceq \frac{1}{2(k + 1)\eta} \cdot \mathbf{I}.$$

Besides, we also have $\mathbf{\Sigma}(\mathbf{I} - \eta\mathbf{\Sigma})^{2k} \preceq \mathbf{\Sigma}$. This implies that

$$\Theta_1 \leq \min\left\{\mathbf{x}_i^\top\mathbf{\Sigma}\mathbf{x}_i, \frac{\|\mathbf{x}_i\|_2^2}{2(k + 1)\eta}\right\} \leq \min\left\{\|\mathbf{\Sigma}\|_2 \cdot \|\mathbf{x}_i\|_2^2, \frac{\|\mathbf{x}_i\|_2^2}{2(k + 1)\eta}\right\}. \tag{B.1}$$

Then applying Lemma B.2 and using the assumption that $\lambda_1 = \Theta(1)$, we have

$$\|\mathbf{\Sigma}\|_2 \lesssim \|\mathbf{H}\|_2.$$

Besides, by Assumption 3.1, we have

$$\|\mathbf{x}_i\|_2^2 = \sum_i \lambda_i \cdot z_i^2$$

where $z_i$ is independent 1-subgaussian random variable and satisfies $\mathbb{E}[z_i^2] = 1$. Therefore, applying Lemma B.3 we can get with probability $1 - \delta$,

$$\|\mathbf{x}_i\|_2^2 \lesssim \sum_i \lambda_i + \max\left\{\log(1/\delta) \cdot \lambda_1, \sqrt{\log(1/\delta)\sum_i \lambda_i^2}\right\}.$$

Setting $\delta = 1/\text{poly}(n)$ and applying union bound over all $i \in [n]$, we can get with probability at least $1 - 1/\text{poly}(n)$, it holds that $\|\mathbf{x}_i\|_2^2 \leq \log(n) \cdot \text{tr}(\mathbf{H})$ for all $i \in [n]$. Putting this into (B.1) completes the proof.

∎

## B.3 Proof of Lemma 5.3

We first provide the following useful facts and lemmas.

**Fact B.4 (Part of Lemma 8 in Bartlett et al. [4])** *The gram matrix $\mathbf{A} = \mathbf{X}\mathbf{X}^\top$ can be decomposed by*

$$\mathbf{A} = \sum_i \lambda_i\mathbf{z}_i\mathbf{z}_i^\top,$$

*where $\mathbf{z}_i \in \mathbb{R}^n$ are independent 1-subgaussian random vector satisfying $\mathbb{E}[\|\mathbf{z}_i\|_2^2] = n$.*

**Fact B.5** *Assume $n < d$ and the gram matrix $\mathbf{A}$ is of full-rank, then it holds that*

$$\mathbf{X}(\mathbf{I}_d - \eta\mathbf{\Sigma})^k = (\mathbf{I}_n - \eta n^{-1}\mathbf{A})^k\mathbf{X}.$$

**Proof.** [Proof of Fact B.5] Note that $\mathbf{X} \in \mathbb{R}^{n \times d}$, consider its SVD decomposition $\mathbf{X} = \mathbf{U}\mathbf{\Lambda}\mathbf{V}^\top$, where $\mathbf{U} \in \mathbb{R}^{n \times n}$, $\mathbf{V} \in \mathbb{R}^{d \times d}$ and $\mathbf{\Lambda} \in \mathbb{R}^{n \times d}$. Then we have $\mathbf{\Sigma} = n^{-1}\mathbf{X}^\top\mathbf{X} = n^{-1}\mathbf{V}\mathbf{\Lambda}^\top\mathbf{\Lambda}\mathbf{V}^\top$, which implies that

$$\mathbf{X}(\mathbf{I} - \eta\mathbf{\Sigma})^k = \mathbf{U}\mathbf{\Lambda}\mathbf{V}^\top\mathbf{V}(\mathbf{I}_d - \eta n^{-1}\mathbf{\Lambda}^\top\mathbf{\Lambda})^k\mathbf{V}^\top = \mathbf{U}\mathbf{\Lambda}(\mathbf{I}_d - \eta n^{-1}\mathbf{\Lambda}^\top\mathbf{\Lambda})^k\mathbf{V}^\top.$$

Additionally, it is easy to verify that $\mathbf{\Lambda}(\mathbf{I}_d - \eta n^{-1}\mathbf{\Lambda}^\top\mathbf{\Lambda}) = (\mathbf{I}_n - \eta n^{-1}\mathbf{\Lambda}\mathbf{\Lambda}^\top)\mathbf{\Lambda}$. Therefore, it follows that

$$\mathbf{X}(\mathbf{I} - \eta\mathbf{\Sigma})^k = \mathbf{U}\mathbf{\Lambda}(\mathbf{I}_d - \eta n^{-1}\mathbf{\Lambda}^\top\mathbf{\Lambda})^k\mathbf{V}^\top = \mathbf{U}(\mathbf{I}_n - \eta n^{-1}\mathbf{\Lambda}\mathbf{\Lambda}^\top)^k\mathbf{\Lambda}\mathbf{V}^\top = (\mathbf{I}_n - \eta\mathbf{A})^k\mathbf{X},$$

where the last equality follows from the fact that $\mathbf{A} = \mathbf{U}\mathbf{\Lambda}\mathbf{\Lambda}^\top\mathbf{U}^\top$. This completes the proof. ∎

**Lemma B.6** *Let $\mathbf{u} \in \mathcal{S}^{n-1}$ be a uniformly random unit vector, then for any fixed PSD matrix $\mathbf{\Theta} \in \mathbb{R}^{n \times n}$, with probability at least $1 - 1/\mathrm{poly}(n)$, it holds that*

$$\mathbf{u}^\top\mathbf{\Theta}\mathbf{u} \lesssim \frac{\log(n)}{n} \cdot \mathrm{tr}(\mathbf{\Theta}).$$

**Proof.** We first consider a Gaussian random vector $\mathbf{v} \sim N(0, \mathbf{I}_n/n)$, then it is clear that we can reformulate it as $\mathbf{v} = r \cdot \mathbf{u}$, where $\mathbf{u}$ is a uniformly random unit vector and $\mathbb{E}[r] = 1$. Note that $nr^2$ follows $\chi^2(n)$ distribution, then by standard concentration result for sub-exponential random variable [36], we have with probability at least $1 - e^{-cn}$ for some small constant $c > 0$ that $r \geq 1/2$. Moreover, let $\mathbf{\Theta} = \sum_i \mu_i\mathbf{z}_i\mathbf{z}_i^\top$ be the eigen-decomposition of $\mathbf{\Theta}$, we have

$$n\mathbf{v}^\top\mathbf{\Theta}\mathbf{v} - \mathrm{tr}(\mathbf{\Theta}) = \sum_{i=1}^n \mu_i[n(\mathbf{z}_i^\top\mathbf{v})^2 - 1] := \sum_{i=1}^n \mu_i\xi_i$$

where $\xi_i \sim \chi^2(1) - 1$ distribution, which is 1-subexponential. Then applying Lemma B.3, we have with probability at least $1 - 2e^{-x}$ such that

$$\sum_{i=1}^n \mu_i\xi_i \leq C \cdot \max\left(x\mu_1, \sqrt{x\sum_{i=1}^n \mu_i^2}\right)$$

holds for some constant $C$.

Combining the previous results, we have with probability at least $1 - e^{cn} - 2e^{-x}$,

$$\mathbf{u}^\top\mathbf{\Theta}\mathbf{u} = r^{-1}\mathbf{v}^\top\mathbf{\Theta}\mathbf{v} \leq \frac{2}{n}\left[\mathrm{tr}(\mathbf{\Theta}) + C \cdot \max\left(x\mu_1, \sqrt{x\sum_{i=1}^n \mu_i^2}\right)\right].$$

Further note that $\sum_{i=1}^n \mu_i^2, \mu_1 \leq \mathrm{tr}^2(\mathbf{\Theta})$, then setting $x = C'\log(n)$ for some absolute constant $C'$, we have with probability at least $1 - 1/\mathrm{poly}(n)$,

$$\mathbf{u}^\top\mathbf{\Theta}\mathbf{u} = r^{-1}\mathbf{v}^\top\mathbf{\Theta}\mathbf{v} \leq \frac{C''\log(n)}{n} \cdot \mathrm{tr}(\mathbf{\Theta})$$

for some absolute constant $C''$. This completes the proof.

∎

**Lemma B.7** *For any $k^* \in [d]$, with probability at least $1 - 1/\mathrm{poly}(n)$, it holds that*

$$\mathrm{tr}(\mathbf{A}(\mathbf{I}_n - \eta n^{-1}\mathbf{A})^{2k}) \lesssim \frac{nk^*}{(k+1)\eta} + n\log(n) \cdot \sum_{i>k^*} \lambda_i.$$

**Proof.** Let $\mu_1, \ldots, \mu_n$ be the sorted (in descending order) eigenvalues of $\mathbf{A}$, then we have

$$\text{tr}\left(\mathbf{A}(\mathbf{I}_n - \eta n^{-1}\mathbf{A})^{2k}\right)) = \sum_{i=1}^n \mu_i \cdot (1 - \eta n^{-1}\mu_i)^{2k} \le \sum_{i=1}^n \min\left\{\frac{n}{2(k+1)\eta}, \mu_i\right\}, \quad \text{(B.2)}$$

where the inequality follows from the fact that $(1 - x)^k \le 1/[(k+1)x]$ for all $x \in (0,1)$ and $k > 0$. Additionally, by Fact B.4 we have

$$\mathbf{A} = \sum_i \lambda_i \mathbf{z}_i \mathbf{z}_i^\top,$$

where $\{\mathbf{z}_i\}_{i=1,\ldots,n}$ are i.i.d. 1-subgaussian random vectors satisfying $\mathbb{E}[\mathbf{z}_i] = 0$ and $\mathbb{E}[\|\mathbf{z}_i\|_2^2] = n$. Then define

$$\mathbf{A}_k := \sum_{i>k} \lambda_i \mathbf{z}_i \mathbf{z}_i^\top, \quad \text{(B.3)}$$

and

$$\mathbf{A}_k = \sum_{i=1}^n \mu_i(\mathbf{A}_k) \mathbf{u}_i \mathbf{u}_i^\top$$

be its eigen-decomposition. Then note that $\mathbf{A} - \mathbf{A}_k + \sum_{i=1}^j \mu_i(\mathbf{A}_k) \mathbf{u}_i \mathbf{u}_i^\top$ has rank at most $k + j$, thus there must exist a linear space $\mathcal{L}$ of dimension $n - k - j$ (that is orthogonal to $\{\mathbf{z}_i\}_{i=1,\ldots,k}$ and $\{\mathbf{u}_i\}_{i=1}^j$) such that for all $\mathbf{v} \in \mathcal{L}$,

$$\mathbf{v}^\top \mathbf{A} \mathbf{v} \le \mathbf{v}^\top \mu_1\left(\mathbf{A}_k - \sum_{i=1}^j \mu_i(\mathbf{A}_k) \mathbf{u}_i \mathbf{u}_i^\top\right) \mathbf{v} = \mathbf{v}^\top \mu_{j+1}(\mathbf{A}_k) \mathbf{v}.$$

This implies that for any $k \in [n]$ and $j \in [n - k]$, it holds that

$$\mu_{k+j}(\mathbf{A}) \le \mu_j(\mathbf{A}_k),$$

and thus

$$\sum_{i=k+1}^n \mu_i \le \sum_{i=1}^{n+1-i} \mu_i(\mathbf{A}_k) \le \text{tr}(\mathbf{A}_k). \quad \text{(B.4)}$$

Moreover, by the definition of $\mathbf{A}_k$ in (B.3), we have

$$\text{tr}(\mathbf{A}_k) = \sum_{i>k} \lambda_i \|\mathbf{z}_i\|_2^2.$$

Then note that $\|\mathbf{z}_i\|_2^2/n - 1$ is 1-subexponential, by Lemma B.3, we have with probability at least $1 - 2e^{-x}$

$$\text{tr}(\mathbf{A}_k) \le n \sum_{i>k} \lambda_i + C \cdot n \cdot \max\left(x\lambda_{k+1}, \sqrt{x \sum_{i>k} \lambda_i^2}\right).$$

for some absolute constant $C$. Then setting $x = \Theta(\log(n))$ and using the fact that $\sum_{i>k} \lambda_i^2 \le (\sum_{i>k} \lambda_i)^2$, we have with probability at least $1 - 1/\text{poly}(n)$,

$$\text{tr}(\mathbf{A}_k) \lesssim n \log(n) \cdot \sum_{i>k} \lambda_i. \quad \text{(B.5)}$$

Putting (B.5) into (B.4) and further applying (B.2), we have for any $k^* \in [n]$, with probability at least $1 - 1/\text{poly}(n)$

$$\text{tr}\left(\mathbf{A}(\mathbf{I}_n - \eta n^{-1}\mathbf{A})^{2k}\right)) \le \sum_{i=1}^{k^*} \frac{n}{2(k+1)\eta} + \text{tr}(\mathbf{A}_k) \lesssim \frac{nk^*}{(k+1)\eta} + n \log(n) \cdot \sum_{i>k^*} \lambda_i.$$

This completes the proof. ■

**Proof.** [Proof of Lemma 5.3] Recalling the formula of $\Theta_2$, we have

$$\Theta_2 = \mathbf{x}_i^\top (\mathbf{I} - \eta \boldsymbol{\Sigma})^k (\mathbf{H} - \boldsymbol{\Sigma})(\mathbf{I} - \eta \boldsymbol{\Sigma})^k \mathbf{x}_i.$$

Moreover, note that $\mathbf{x}_i$ can be rewritten as $\mathbf{x}_i = \mathbf{e}_i^\top \mathbf{X}$, where $\mathbf{e}_i \in \mathbb{R}^n$ and $\mathbf{X} \in \mathbb{R}^{n \times d}$. Then

$$\begin{aligned}
\Theta_2 &= \mathbf{e}_i^\top \mathbf{X}(\mathbf{I} - \eta \boldsymbol{\Sigma})^k (\mathbf{H} - \boldsymbol{\Sigma})(\mathbf{I} - \eta \boldsymbol{\Sigma})^k \mathbf{X}^\top \mathbf{e}_i \\
&\leq \|\mathbf{e}_i^\top \mathbf{X}(\mathbf{I} - \eta \boldsymbol{\Sigma})^k\|_2^2 \cdot \|\mathbf{H} - \boldsymbol{\Sigma}\|_2.
\end{aligned} \tag{B.6}$$

Then by Fact B.5, we have

$$\begin{aligned}
\|\mathbf{e}_i^\top \mathbf{X}(\mathbf{I} - \eta \boldsymbol{\Sigma})^k\|_2^2 &= \|\mathbf{e}_i^\top (\mathbf{I}_n - \eta n^{-1} \mathbf{A})^k \mathbf{X}\| \\
&= \mathbf{e}_i^\top (\mathbf{I}_n - \eta n^{-1} \mathbf{A})^k \mathbf{X} \mathbf{X}^\top (\mathbf{I}_n - \eta n^{-1} \mathbf{A})^k \mathbf{e}_i \\
&= \mathbf{e}_i^\top \mathbf{A}(\mathbf{I}_n - \eta n^{-1} \mathbf{A})^{2k} \mathbf{e}_i.
\end{aligned}$$

Note that $\mathbf{e}_i$ is independent of the randomness of $\mathbf{A}$ and the eigenvectors of $\mathbf{A}$ is rotation invariant. Specifically, note that $\mathbf{A} = \mathbf{U}\boldsymbol{\Lambda}\boldsymbol{\Lambda}^\top \mathbf{U}^\top$, where $\mathbf{U} \in \mathbb{R}^{n \times n}$ is an orthonormal matrix and $\boldsymbol{\Lambda}\boldsymbol{\Lambda}^\top \in \mathbb{R}^{n \times n}$ is an diagonal matrix. Then we consider the conditional distribution $\mathbb{P}(\mathbf{A}|\boldsymbol{\Lambda}\boldsymbol{\Lambda}^\top)$, which can be viewed as a distribution over the orthonormal matrix $\mathbf{U}$, denoted by $\mathbb{P}(\mathbf{U})$. Then note that $\mathbf{U}$ can also be understood as a rotation matrix when operated on an vector, and using Fact B.4, we have for any rotation matrix $\mathbf{P}$, it holds that

$$\mathbf{P}\mathbf{A}\mathbf{P}^\top = \sum_i \lambda_i \mathbf{P}\mathbf{z}_i \mathbf{z}_i^\top \mathbf{P}^\top$$

which has the same distribution of $\mathbf{A} = \sum_i \lambda_i \mathbf{z}_i \mathbf{z}_i^\top$ since $\mathbf{P}\mathbf{z}_i$ and $\mathbf{z}_i$ have the same distribution. Therefore, it can be verified that for any different orthonormal matrices $\mathbf{U}_1$ and $\mathbf{U}_2$ and let $\mathbf{P} = \mathbf{U}_2 \mathbf{U}_1^\top$, which is also an orthonormal matrix, we have

$$\mathbb{P}(\mathbf{U}_1 \boldsymbol{\Lambda}\boldsymbol{\Lambda}^\top \mathbf{U}_1^\top | \boldsymbol{\Lambda}\boldsymbol{\Lambda}^\top) = \mathbb{P}(\mathbf{P}\mathbf{U}_1 \boldsymbol{\Lambda}\boldsymbol{\Lambda}^\top \mathbf{U}_1^\top \mathbf{P}^\top | \boldsymbol{\Lambda}\boldsymbol{\Lambda}^\top) = \mathbb{P}(\mathbf{U}_2 \boldsymbol{\Lambda}\boldsymbol{\Lambda}^\top \mathbf{U}_2^\top | \boldsymbol{\Lambda}\boldsymbol{\Lambda}^\top).$$

This implies that $\mathbb{P}(\mathbf{U}_1) = \mathbb{P}(\mathbf{U}_2)$ for any $\mathbf{U}_1 \neq \mathbf{U}_2$. Therefore, we can conclude that $\mathbb{P}(\mathbf{U})$ is an uniform distribution over the entire class of orthonormal matrices. Then note that

$$\mathbf{A}(\mathbf{I}_n - \eta n^{-1} \mathbf{A})^{2k} = \mathbf{P}\big(\boldsymbol{\Lambda}\boldsymbol{\Lambda}^\top (\mathbf{I} - n^{-1} \eta \boldsymbol{\Lambda}\boldsymbol{\Lambda}^\top)^{2k}\big)\mathbf{P}^\top.$$

Then for any fixed $i$, using the fact that $\mathbf{P}$ is a uniformly random rotation matrix, we have $\mathbf{P}^\top \mathbf{e}_i$ is a random unit vector in $\mathcal{S}^{n-1}$. Then applying Lemmas B.6 and B.7, and taking union bound over $i \in [n]$, we have with probability at least $1 - 1/\mathrm{poly}(n)$,

$$\begin{aligned}
\mathbf{e}_i^\top \mathbf{A}(\mathbf{I}_n - \eta n^{-1} \mathbf{A})^{2k} \mathbf{e}_i &\lesssim \frac{\log(n)}{n} \cdot \mathrm{tr}\big(\mathbf{A}(\mathbf{I}_n - \eta n^{-1} \mathbf{A})^{2k}\big) \\
&\lesssim \log(n) \cdot \left(\frac{k^*}{(k+1)\eta} + \log(n) \cdot \sum_{i > k^*} \lambda_i\right).
\end{aligned} \tag{B.7}$$

Finally, applying Lemma B.2 and setting $\delta = 1/\mathrm{poly}(n)$, we have

$$\|\mathbf{H} - \boldsymbol{\Sigma}\|_2 \lesssim \sqrt{\frac{\log(n)}{n}}. \tag{B.8}$$

Putting (B.8) and (B.7) into (B.6), we can obtain

$$\Theta_2 \leq \|\mathbf{e}_i^\top \mathbf{X}(\mathbf{I} - \eta \boldsymbol{\Sigma})^k\|_2^2 \cdot \|\mathbf{H} - \boldsymbol{\Sigma}\|_2 \lesssim \frac{\log^{5/2}(n)}{n^{1/2}} \cdot \left(\frac{k^*}{(k+1)\eta} + \sum_{i > k^*} \lambda_i\right),$$

which completes the proof. $\blacksquare$

## B.4 Completing the analysis for fluctuation error: Proof of Theorem 4.2

Combining the established upper bounds on $\Theta_1$ and $\Theta_2$ in Lemmas 5.2 and 5.3 gives the following lemma.

**Lemma B.8** *If the stepsize satisfies $\gamma \leq 1/(c\operatorname{tr}(\mathbf{H}))$ for some absolute constant $c$, then with probability at least $1 - 1/\operatorname{poly}(n)$, there exists an absolute constant $C$ such that*

$$\mathcal{M} \circ \mathcal{G}^k \circ \mathbf{H} \preceq C \cdot \left[ \log(n) \cdot \min\left\{ \frac{1}{(k+1)\eta}, \|\mathbf{H}\|_2 \right\} \cdot \operatorname{tr}(\mathbf{H}) + \frac{\log^{5/2}(n)}{n^{1/2}} \cdot \left( \frac{k^*}{(k+1)\eta} + \sum_{i>k^*} \lambda_i \right) \right] \cdot \boldsymbol{\Sigma}.$$

**Lemma B.9** *For any $t > 0$, if the stepsize satisfies $\eta \leq 1/(c\operatorname{tr}(\mathbf{H})\log(t))$ for some absolute constant $c$, then it holds that*

$$\sum_{k=0}^{t-1} \langle \boldsymbol{\Sigma}, \mathbf{E}_k \rangle \lesssim \frac{1}{\eta} \cdot \langle \mathbf{I} - (\mathbf{I} - \eta\boldsymbol{\Sigma})^t, \mathbf{E}_0 \rangle,$$

$$\sum_{k=0}^{t-1} \frac{\langle \boldsymbol{\Sigma}, \mathbf{E}_k \rangle}{t - k} \lesssim \frac{1}{\eta t} \langle (\mathbf{I} - (\mathbf{I} - \eta\boldsymbol{\Sigma})^t), \mathbf{E}_0 \rangle + \log(t) \langle (\mathbf{I} - \eta\boldsymbol{\Sigma})^t \boldsymbol{\Sigma}, \mathbf{E}_0 \rangle.$$

**Proof.** [Proof of Lemma B.9] In this part we seek to bound $\sum_{k=0}^{t-1} \langle \boldsymbol{\Sigma}, \mathbf{E}_k \rangle$ and $\sum_{k=0}^{t-1} \frac{\langle \boldsymbol{\Sigma}, \mathbf{E}_k \rangle}{t-k}$ in separate. By (5.2), we can get

$$\langle \boldsymbol{\Sigma}, \mathbf{E}_t \rangle \leq \langle \boldsymbol{\Sigma}, \mathcal{G}^t \circ \mathbf{E}_0 \rangle + \eta^2 \sum_{k=0}^{t-1} \langle \boldsymbol{\Sigma}, \mathcal{G}^{t-1-k} \circ \mathcal{M} \circ \mathbf{E}_k \rangle$$

$$= \langle \mathcal{G}^t \circ \boldsymbol{\Sigma}, \mathbf{E}_0 \rangle + \eta^2 \sum_{k=0}^{t-1} \langle \mathcal{M} \circ \mathcal{G}^{t-1-k} \circ \boldsymbol{\Sigma}, \mathbf{E}_k \rangle$$

$$= \langle (\mathbf{I} - \eta\boldsymbol{\Sigma})^{2t} \boldsymbol{\Sigma}, \mathbf{E}_0 \rangle + \eta^2 \sum_{k=0}^{t-1} \langle \mathcal{M} \circ \left( (\mathbf{I} - \eta\boldsymbol{\Sigma})^{2(t-1-k)} \boldsymbol{\Sigma} \right), \mathbf{E}_k \rangle. \tag{B.9}$$

Note that $(\mathbf{I} - \eta\boldsymbol{\Sigma})^{2(t-1-k)} \boldsymbol{\Sigma} \preceq \frac{1}{\eta(t-k)} \mathbf{I}$, and $\mathcal{M} \circ \mathbf{I} \preceq c\operatorname{tr}(\mathbf{H})\boldsymbol{\Sigma}$ for some absolute constant $c$, we then have the following by (B.9)

$$\langle \boldsymbol{\Sigma}, \mathbf{E}_t \rangle \leq \langle (\mathbf{I} - \eta\boldsymbol{\Sigma})^{2t} \boldsymbol{\Sigma}, \mathbf{E}_0 \rangle + c\eta \operatorname{tr}(\mathbf{H}) \sum_{k=0}^{t-1} \frac{\langle \boldsymbol{\Sigma}, \mathbf{E}_k \rangle}{t - k}. \tag{B.10}$$

We now bound $\sum_{k=0}^{t-1} \langle \boldsymbol{\Sigma}, \mathbf{E}_k \rangle$ by recursively applying (B.10) to establish

$$\sum_{k=0}^{t-1} \langle \boldsymbol{\Sigma}, \mathbf{E}_k \rangle \leq \langle \sum_{k=0}^{t-1} (\mathbf{I} - \eta\boldsymbol{\Sigma})^{2k} \boldsymbol{\Sigma}, \mathbf{E}_0 \rangle + c\eta \operatorname{tr}(\mathbf{H}) \sum_{k=0}^{t-1} \sum_{i=0}^{k-1} \frac{\langle \boldsymbol{\Sigma}, \mathbf{E}_i \rangle}{k - i}$$

$$\leq \frac{1}{\eta} \langle \mathbf{I} - (\mathbf{I} - \eta\boldsymbol{\Sigma})^t, \mathbf{E}_0 \rangle + 2c\eta \operatorname{tr}(\mathbf{H}) \log(t) \sum_{i=0}^{t-1} \langle \boldsymbol{\Sigma}, \mathbf{E}_i \rangle,$$

and conclude that

$$\sum_{k=0}^{t-1} \langle \boldsymbol{\Sigma}, \mathbf{E}_k \rangle \leq \frac{1}{1 - 2c\eta \operatorname{tr}(\mathbf{H})\log(t)} \cdot \frac{1}{\eta} \cdot \langle \mathbf{I} - (\mathbf{I} - \eta\boldsymbol{\Sigma})^t, \mathbf{E}_0 \rangle \tag{B.11}$$

$$\leq C \cdot \frac{1}{\eta} \cdot \langle \mathbf{I} - (\mathbf{I} - \eta\boldsymbol{\Sigma})^t, \mathbf{E}_0 \rangle. \tag{B.12}$$

Similarly, we then bound $\sum_{k=0}^{t-1} \frac{\langle \boldsymbol{\Sigma}, \mathbf{E}_k \rangle}{t-k}$ by recursively applying (B.10) to establish

$$\sum_{k=0}^{t-1} \frac{\langle \boldsymbol{\Sigma}, \mathbf{E}_k \rangle}{t - k} \leq \langle \sum_{k=0}^{t-1} \frac{(\mathbf{I} - \eta\boldsymbol{\Sigma})^{2k} \boldsymbol{\Sigma}}{t - k}, \mathbf{E}_0 \rangle + c\eta \operatorname{tr}(\mathbf{H}) \sum_{k=0}^{t-1} \sum_{i=0}^{k-1} \frac{\langle \boldsymbol{\Sigma}, \mathbf{E}_i \rangle}{(t-k)(k-i)}$$

$$\leq \langle \sum_{k=0}^{t-1} \frac{(\mathbf{I} - \eta\boldsymbol{\Sigma})^{2k}\boldsymbol{\Sigma}}{t-k}, \mathbf{E}_0\rangle + 2c\eta\,\mathrm{tr}(\mathbf{H})\log(t)\sum_{i=0}^{t-1}\frac{\langle\boldsymbol{\Sigma}, \mathbf{E}_i\rangle}{t-i},$$

so we can conclude that

$$\sum_{k=0}^{t-1}\frac{\langle\boldsymbol{\Sigma}, \mathbf{E}_k\rangle}{t-k} \leq \frac{1}{1 - 2c\eta\,\mathrm{tr}(\mathbf{H})\log(t)}\langle\sum_{k=0}^{t-1}\frac{(\mathbf{I} - \eta\boldsymbol{\Sigma})^{2k}\boldsymbol{\Sigma}}{t-k}, \mathbf{E}_0\rangle \tag{B.13}$$

$$\lesssim \sum_{k=0}^{t-1}\frac{(\mathbf{I} - \eta\boldsymbol{\Sigma})^{2k}\boldsymbol{\Sigma}}{t-k}, \mathbf{E}_0\rangle \tag{B.14}$$

$$\lesssim \left(\frac{1}{\eta t}\langle(\mathbf{I} - (\mathbf{I} - \eta\boldsymbol{\Sigma})^t), \mathbf{E}_0\rangle + \log(t)\langle(\mathbf{I} - \eta\boldsymbol{\Sigma})^t\boldsymbol{\Sigma}, \mathbf{E}_0\rangle\right), \tag{B.15}$$

where the last inequality is due to

$$\sum_{k=0}^{t-1}\frac{(\mathbf{I} - \eta\boldsymbol{\Sigma})^{2k}\boldsymbol{\Sigma}}{t-k} \lesssim \frac{1}{\eta t}(\mathbf{I} - (\mathbf{I} - \eta\boldsymbol{\Sigma})^t) + \log(t)\cdot(\mathbf{I} - \eta\boldsymbol{\Sigma})^t\boldsymbol{\Sigma}.$$

∎

**Lemma B.10** *For any $t \geq 0$ and $\eta \leq 1/(c\,\mathrm{tr}(\mathbf{H})\log(t))$ for some absolute constant c, it holds that*

$$\langle\mathbf{I} - (\mathbf{I} - \eta\boldsymbol{\Sigma})^t, \mathbf{E}_0\rangle \leq \min\left\{\|\widehat{\mathbf{w}}\|_2^2, t\eta\cdot\langle\boldsymbol{\Sigma}, \mathbf{E}_0\rangle\right\}$$
$$t\eta\cdot\langle(\mathbf{I} - \eta\boldsymbol{\Sigma})^t\boldsymbol{\Sigma}, \mathbf{E}_0\rangle \leq \min\left\{\|\widehat{\mathbf{w}}\|_2^2, t\eta\cdot\langle\boldsymbol{\Sigma}, \mathbf{E}_0\rangle\right\}$$

**Proof.** According to the definition of $\mathbf{E}_t$ and applying zero initialization $\mathbf{w}_0 = \mathbf{0}$, then we have $\mathbf{E}_0 = \widehat{\mathbf{w}}\widehat{\mathbf{w}}^\top \preceq \|\widehat{\mathbf{w}}\|_2^2\cdot\mathbf{I}$. Moreover, note that our choice of stepsize guarantees that $\mathbf{I} - \eta\boldsymbol{\Sigma}$ is a PSD matrix, we have

$$\mathbf{I} - (\mathbf{I} - \eta\boldsymbol{\Sigma})^t \preceq \mathbf{I}, \quad \mathbf{I} - (\mathbf{I} - \eta\boldsymbol{\Sigma})^t \preceq t\eta\boldsymbol{\Sigma}, \quad (\mathbf{I} - \eta\boldsymbol{\Sigma})^t\boldsymbol{\Sigma} \preceq \boldsymbol{\Sigma}, \quad (\mathbf{I} - \eta\boldsymbol{\Sigma})^t\boldsymbol{\Sigma} \preceq \frac{1}{t\eta}\cdot\mathbf{I}.$$

Then it follows that

$$\langle\mathbf{I} - (\mathbf{I} - \eta\boldsymbol{\Sigma})^t, \mathbf{E}_0\rangle \leq \min\left\{\langle\mathbf{I}, \mathbf{E}_0\rangle, t\eta\cdot\langle\boldsymbol{\Sigma}, \mathbf{E}_0\rangle\right\} = \min\left\{\|\widehat{\mathbf{w}}\|_2^2, t\eta\cdot\langle\boldsymbol{\Sigma}, \mathbf{E}_0\rangle\right\}$$
$$\langle(\mathbf{I} - \eta\boldsymbol{\Sigma})^t\boldsymbol{\Sigma}, \mathbf{E}_0\rangle \leq \min\left\{\langle\boldsymbol{\Sigma}, \mathbf{E}_0\rangle, \frac{1}{t\eta}\cdot\langle\mathbf{I}, \mathbf{E}_0\rangle\right\} = \min\left\{\langle\boldsymbol{\Sigma}, \mathbf{E}_0\rangle, \frac{\|\widehat{\mathbf{w}}\|_2^2}{t\eta}\right\}.$$

This completes the proof.

∎

Now we are ready to complete the proof of Theorem 4.2. **Proof.** [Proof of Theorem 4.2] By Lemma B.1, we have

$$\underbrace{\mathrm{FluctuationError}}_{*} \leq \frac{\eta^2}{2}\cdot\sum_{k=0}^{t-1}\langle\mathcal{M}\circ\mathcal{G}^{t-1-k}\circ\mathbf{H}, \mathbf{E}_k\rangle.$$

Additionally, by Lemma B.8, we further have

$$(*) \lesssim \eta^2\cdot\sum_{k=0}^{t-1}\left[\frac{\log(n)}{(t-k)\eta}\cdot\mathrm{tr}(\mathbf{H}) + \frac{\log^{5/2}(n)}{n^{1/2}}\cdot\left(\frac{k^*}{(t-k)\eta} + \sum_{i>k^*}\lambda_i\right)\right]\cdot\langle\boldsymbol{\Sigma}, \mathbf{E}_k\rangle$$

$$\lesssim \eta\cdot\left(\log(n)\,\mathrm{tr}(\mathbf{H}) + \frac{k^*\log^{5/2}(n)}{n^{1/2}}\right)\cdot\sum_{k=0}^{t-1}\frac{\langle\boldsymbol{\Sigma}, \mathbf{E}_k\rangle}{t-k} + \eta^2\cdot\frac{\log^{5/2}(n)}{n^{1/2}}\cdot\sum_{i>k^*}\lambda_i\cdot\sum_{k=0}^{t-1}\langle\boldsymbol{\Sigma}, \mathbf{E}_k\rangle.$$

Then applying Lemma B.9, we can further obtain

$$(*) \lesssim \eta\cdot\left(\log(n)\,\mathrm{tr}(\mathbf{H}) + \frac{k^*\log^{5/2}(n)}{n^{1/2}}\right)\cdot\left(\frac{1}{\eta t}\langle(\mathbf{I} - (\mathbf{I} - \eta\boldsymbol{\Sigma})^t), \mathbf{E}_0\rangle + \log(t)\langle(\mathbf{I} - \eta\boldsymbol{\Sigma})^t\boldsymbol{\Sigma}, \mathbf{E}_0\rangle\right)$$

$$+ \eta \cdot \frac{\log^{5/2}(n)}{n^{1/2}} \cdot \sum_{i>k^*} \lambda_i \cdot \langle \mathbf{I} - (\mathbf{I} - \eta\mathbf{\Sigma})^t, \mathbf{E}_0 \rangle$$

$$= \left( \frac{\log(n)\operatorname{tr}(\mathbf{H})}{t} + \frac{\log^{5/2}(n)}{n^{1/2}t} \cdot \left(k^* + \eta t \sum_{i>k^*} \lambda_i\right) \right) \cdot \langle (\mathbf{I} - (\mathbf{I} - \eta\mathbf{\Sigma})^t), \mathbf{E}_0 \rangle$$

$$+ \eta \log(t) \cdot \left( \log(n)\operatorname{tr}(\mathbf{H}) + \frac{k^* \log^{5/2}(n)}{n^{1/2}} \right) \cdot \langle (\mathbf{I} - \eta\mathbf{\Sigma})^t \mathbf{\Sigma}, \mathbf{E}_0 \rangle$$

$$\lesssim \left[ \log(t) \cdot \left( \frac{\operatorname{tr}(\mathbf{H})\log(n)}{t} + \frac{k^* \log^{5/2}(n)}{n^{1/2}t} \right) + \frac{\log^{5/2}(n)\eta}{n^{1/2}} \cdot \sum_{i>k^*} \lambda_i \right] \cdot \min\left\{ \|\widehat{\mathbf{w}}\|_2^2, t\eta \cdot \langle \mathbf{\Sigma}, \mathbf{E}_0 \rangle \right\}.$$

where the last inequality follows from Lemma B.10.

∎

# C  Risk bounds for Gradient Descent with Early Stopping

## C.1  Proof of Lemma A.1

**Proof.** [Proof of Lemma A.1] For the first inequality, note that

$$\mathbf{I} - (\mathbf{I} - \eta n^{-1}\mathbf{A})^t \preceq \begin{cases} \mathbf{I}; \\ n^{-1}\eta t\mathbf{A}, \end{cases}$$

we then obtain

$$\widetilde{\mathbf{A}} := \mathbf{A}\big(\mathbf{I} - (\mathbf{I} - \eta n^{-1}\mathbf{A})^t\big)^{-1} \succeq \begin{cases} \mathbf{A}; \\ \frac{n}{\eta t}\mathbf{I}. \end{cases}$$

Therefore

$$\widetilde{\mathbf{A}} \succeq \frac{1}{2}\big(\mathbf{A} + \frac{n}{\eta t}\mathbf{I}\big).$$

For the second inequality, note that

$$\widetilde{\mathbf{A}} - \mathbf{A} = \mathbf{A}(\mathbf{I} - \eta n^{-1}\mathbf{A})^t \big[\mathbf{I} - (\mathbf{I} - \eta n^{-1}\mathbf{A})^t\big]^{-1}.$$

Then it suffices to consider the scalar function $f(x) := nx(1 - \eta x)^t / \big[1 - (1 - \eta x)^t\big]$. Then we consider two cases: (1) $t\eta x \geq \log(2)$ and (2) $t\eta x < \log(2)$. For the first case, it is clear that

$$\frac{nx(1 - \eta x)^t}{1 - (1 - \eta x)^t} \leq \frac{n \cdot 1/(t\eta)}{1 - 1/2} = \frac{2n}{t\eta},$$

where we use the inequality $(1 - \eta x)^t x \leq 1/(t\eta)$ in the first inequality. For the case of $t\eta x < \log(2)$, we have $(1 - \eta x)^t \leq 1 - \eta xt/2$ and thus

$$\frac{nx(1 - \eta x)^t}{1 - (1 - \eta x)^t} \leq \frac{nx}{\eta xt/2} = \frac{2n}{t\eta}.$$

Combining the about results in two cases, we have $f(x) \leq 2n/(t\eta)$ and thus

$$\widetilde{\mathbf{A}} = \mathbf{A} + \mathbf{A}(\mathbf{I} - \eta n^{-1}\mathbf{A})^t \big[\mathbf{I} - (\mathbf{I} - \eta n^{-1}\mathbf{A})^t\big]^{-1} \preceq \mathbf{A} + \frac{2n}{t\eta} \cdot \mathbf{I}.$$

This completes the proof of the second inequality.

∎

Then, the lower bound of $\widetilde{\mathbf{A}}$ will be applied to prove the upper bound of variance error of GD, as shown in (A.1), which is at most four times the variance error achieved by the ridge regression with $\lambda = n/(\eta t)$. The upper bound of $\widetilde{\mathbf{A}}$ will be applied to prove the upper bound of the bias error of GD, which is at most the bias error achieved by ridge regression with $\lambda = 2n/(\eta t)$. Finally, we can apply the prior work [33, Theorem 1] on the excess risk analysis for ridge regression to complete the proof for bounding the bias and variance errors separately. The detailed proofs are provided as follows.

## C.2 Variance Error

**Lemma C.1** *For any stepsize $\gamma \le c/\operatorname{tr}(\mathbf{H})$ for some absolute constant $c$ and any $k^* \in [d]$, with probability at least $1 - 1/\operatorname{poly}(n)$,*

$$\mathbb{E}_{\boldsymbol{\epsilon}}[\mathrm{VarError}] \lesssim \frac{k^*}{n} + \frac{n}{\left(n/(\eta t) + \sum_{i>k^*} \lambda_i\right)^2} \cdot \sum_{i>k^*} \lambda_i^2$$

**Proof.** By (A.1), we have

$$\mathbb{E}_{\boldsymbol{\epsilon}}[\mathrm{VarError}] := \left\|\mathbf{X}^\top \widetilde{\mathbf{A}}^{-1} \boldsymbol{\epsilon}\right\|_{\mathbf{H}}^2 \lesssim \operatorname{tr}\left(\mathbf{X}\mathbf{H}\mathbf{X}^\top \widetilde{\mathbf{A}}^{-2}\right) \lesssim \operatorname{tr}\left(\mathbf{X}\mathbf{H}\mathbf{X}^\top \left(\mathbf{A} + \frac{n}{\eta t}\mathbf{I}\right)^{-2}\right), \quad \text{(C.1)}$$

where the last inequality is by Lemma A.1. One finds that (C.1) corresponds to the variance error of ridge regression in [33] for $\lambda = \frac{n}{\eta t}$. Then by Theorem 1 in Tsigler and Bartlett [33], one immediately obtains a bound for GD variance error:

$$\mathbb{E}_{\boldsymbol{\epsilon}}[\mathrm{VarError}] \lesssim \frac{k^*}{n} + \frac{n}{\left(n/(\eta t) + \sum_{i>k^*} \lambda_i\right)^2} \cdot \sum_{i>k^*} \lambda_i^2,$$

where

$$k^* := \min\left\{k : n\lambda_{k+1} \le \frac{n}{\eta t} + \sum_{i>k} \lambda_i\right\}.$$

Setting $\widetilde{\lambda} = n/(\eta t) + \sum_{i>k^*} \lambda_i$ completes the proof. ∎

## C.3 Bias Error

**Lemma C.2** *Assume the ground truth $\mathbf{w}^*$ follows a Gaussian Prior $\mathbf{w}^* \sim \mathcal{N}(0, \omega^2 \cdot \mathbf{I})$. Then for any stepsize $\gamma \le c/\operatorname{tr}(\mathbf{H})$ for some absolute constant $c$ and any $k^* \in [d]$, with probability at least $1 - 1/\operatorname{poly}(n)$,*

$$\mathbb{E}_{\mathbf{w}^*}[\mathrm{BiasError}] \lesssim \omega^2 \cdot \left(\frac{\widetilde{\lambda}^2}{n^2} \cdot \sum_{i \le k^*} \frac{1}{\lambda_i} + \sum_{i>k^*} \lambda_i\right).$$

**Proof.** Note that given the ground truth $\mathbf{w}^*$, the bias error is

$$\mathrm{BiasError} := \|\mathbf{H}^{\frac{1}{2}}\left(\mathbf{I} - \mathbf{X}^\top \widetilde{\mathbf{A}}^{-1}\mathbf{X}\right)\mathbf{w}^*\|_2^2.$$

Further note that

$$\mathbf{w}^* \sim \mathcal{N}(0, \omega^2 \cdot \mathbf{I}_d),$$

then taking expectation over $\mathbf{w}^*$ gives

$$\begin{aligned}
\mathbb{E}_{\mathbf{w}^*}[\mathrm{BiasError}] &= \mathbb{E}_{\mathbf{w}^*}\left[\|\mathbf{H}^{\frac{1}{2}}\left(\mathbf{I} - \mathbf{X}^\top \widetilde{\mathbf{A}}^{-1}\mathbf{X}\right)\mathbf{w}^*\|_2^2\right] \\
&= \omega^2 \cdot \operatorname{tr}\left(\mathbf{H}\left(\mathbf{I} - \mathbf{X}^\top \widetilde{\mathbf{A}}^{-1}\mathbf{X}\right)^2\right) \\
&\le \omega^2 \cdot \underbrace{\operatorname{tr}\left(\mathbf{H}\left(\mathbf{I} - \mathbf{X}^\top \left(\mathbf{A} + \frac{2n}{t\eta}\right)^{-2}\mathbf{X}\right)^2\right)}_{*}
\end{aligned}$$

where the last inequality is by Lemma A.1 and that $\mathbf{A}$ commutes with $\widetilde{\mathbf{A}}$. Moreover, note that the quantity $(*)$ is actually the expected bias error of the ridge regression solution with the regularization parameter $2n/(t\eta)$. Therefore, by Theorem 1 in Tsigler and Bartlett [33], we have

$$\begin{aligned}
(*) &\lesssim \mathbb{E}_{\mathbf{w}^* \sim \mathcal{N}(\mathbf{0},\mathbf{I})}\left[\left(\frac{2n/(\eta t) + \sum_{i>k^*} \lambda_i}{n}\right)^2 \cdot \|\mathbf{w}_{0:k^*}^*\|_{\mathbf{H}_{0:k^*}^{-1}}^2 + \|\mathbf{w}_{k^*:\infty}^*\|_{\mathbf{H}_{k^*:\infty}}^2\right] \\
&\asymp \frac{\widetilde{\lambda}^2}{n^2} \cdot \sum_{i \le k^*} \frac{1}{\lambda_i} + \sum_{i>k^*} \lambda_i,
\end{aligned}$$

where

$$k^* := \min\left\{k : n\lambda_{k+1} \le \frac{n}{\eta t} + \sum_{i>k} \lambda_i\right\},$$

and $\widetilde{\lambda} = n/(\eta t) + \sum_{i>k^*} \lambda_i$. This completes the proof. ∎

## C.4 Proof of Theorem 4.3

**Proof.** [Proof of Theorem 4.3] The proof can be completed by combining Lemmas C.1 and C.2. ∎

# D Proof of Corollaries

## D.1 Proof of Corollary 4.4

The following lemma will be useful in the proof.

**Lemma D.1** *Assume* $\mathbf{w}^* \sim \mathcal{N}(\mathbf{0}, \omega^2 \cdot \mathbf{I})$ *and* $\mathbf{w}_0 = \mathbf{0}$*, then*

$$\mathbb{E}_{\mathbf{w}^*,\boldsymbol{\epsilon}}[\langle \mathbf{E}_0, \boldsymbol{\Sigma} \rangle] \lesssim \omega^2 \cdot \log(n) \cdot \mathrm{tr}(\mathbf{H}) + \sigma^2.$$
$$\mathbb{E}_{\mathbf{w}^*,\boldsymbol{\epsilon}}[\|\widehat{\mathbf{w}}\|_2^2] = n\omega^2 + \sigma^2 \, \mathrm{tr}(\mathbf{A}^{-1}).$$

**Proof.** Applying the formula of $\widehat{\mathbf{w}}$ and the initialization $\mathbf{w}_0 = \mathbf{0}$, we have

$$\langle \mathbf{E}_0, \boldsymbol{\Sigma} \rangle = \langle \mathbf{X}^\top \mathbf{A}^{-1} \mathbf{y} (\mathbf{X}^\top \mathbf{A}^{-1}\mathbf{y})^\top, \boldsymbol{\Sigma} \rangle = \frac{1}{n}\|\mathbf{y}\|_2^2 = \frac{1}{n}\|\mathbf{X}\mathbf{w}^* + \boldsymbol{\epsilon}\|_2^2 \leq \frac{2}{n}\|\mathbf{X}\mathbf{w}^*\|_2^2 + \frac{2}{n}\|\boldsymbol{\epsilon}\|_2^2,$$

where the last inequality follows from Young's inequality. Note that $\boldsymbol{\epsilon}$ is a combination of $n$ independent random variables with variance $\sigma^2$, we have $\mathbb{E}[\|\boldsymbol{\epsilon}\|_2^2] = n\sigma^2$. Besides, regarding the first term, we have with probability at least $1 - 1/\mathrm{poly}(n)$,

$$\mathbb{E}[\|\mathbf{X}\mathbf{w}^*\|_2^2] = \omega^2 \cdot \mathrm{tr}(\mathbf{X}\mathbf{X}^\top) \lesssim \omega^2 \cdot n \cdot \mathrm{tr}(\mathbf{H}).$$

Combining the above results immediately gives

$$\mathbb{E}_{\mathbf{w}^*,\boldsymbol{\epsilon}}[\langle \mathbf{E}_0, \boldsymbol{\Sigma} \rangle] \lesssim \omega^2 \cdot \mathrm{tr}(\mathbf{H}) + \sigma^2.$$

Moreover, note that $\widehat{\mathbf{w}} = \mathbf{X}^\top \mathbf{A}^{-1}\mathbf{y} = \mathbf{X}^\top \mathbf{A}^{-1}(\mathbf{X}\mathbf{w}^* + \boldsymbol{\epsilon})$, we have

$$\begin{aligned}
\mathbb{E}_{\mathbf{w}^*,\boldsymbol{\epsilon}}[\|\widehat{\mathbf{w}}\|_2^2] &= \mathbb{E}_{\mathbf{w}^*,\boldsymbol{\epsilon}}\big[\, \mathrm{tr}\left(\mathbf{X}^\top \mathbf{A}^{-1}(\mathbf{X}\mathbf{w}^* + \boldsymbol{\epsilon})(\mathbf{X}\mathbf{w}^* + \boldsymbol{\epsilon})^\top \mathbf{A}^{-1}\mathbf{X}\right)\big] \\
&= \mathbb{E}_{\mathbf{w}^*,\boldsymbol{\epsilon}}\big[\, \mathrm{tr}\left(\mathbf{X}^\top \mathbf{A}^{-1}\mathbf{X}\mathbf{w}^*\mathbf{w}^{*\top}\mathbf{X}^\top \mathbf{A}^{-1}\mathbf{X}\right) + \mathrm{tr}\left(\mathbf{X}^\top \mathbf{A}^{-1}\boldsymbol{\epsilon}\boldsymbol{\epsilon}^\top \mathbf{A}^{-1}\mathbf{X}\right)\big] \\
&= \omega^2 \, \mathrm{tr}\left(\mathbf{X}^\top \mathbf{A}^{-1}\mathbf{X}\mathbf{X}^\top \mathbf{A}^{-1}\mathbf{X}\right) + \sigma^2 \, \mathrm{tr}(\mathbf{X}^\top \mathbf{A}^{-2}\mathbf{X}),
\end{aligned}$$

where the last equality is due to $\mathbf{w}^* \sim \mathcal{N}(\mathbf{0}, \omega^2\mathbf{I})$ and $\epsilon \sim \mathcal{N}(0, \sigma^2)$. Then note that $\mathbf{A} = \mathbf{X}\mathbf{X}^\top$, we have

$$\mathbb{E}_{\mathbf{w}^*,\boldsymbol{\epsilon}}[\|\widehat{\mathbf{w}}\|_2^2] = \omega^2 \, \mathrm{tr}(\mathbf{I}_n) + \sigma^2 \, \mathrm{tr}(\mathbf{A}^{-1}) = n\omega^2 + \sigma^2 \, \mathrm{tr}(\mathbf{A}^{-1}).$$

This completes the proof.

∎

**Proof.** [Proof of Corollary 4.4] Plugging Lemma D.1 into Theorem 4.2 and then combining Theorems 4.2 and 4.3 completes the proof.

∎

## D.2 Proof of Corollary 4.5

**Proof.** [Proof of Corollary 4.5] First, note that $k^\dagger$ can be arbitrarily chosen, we will first pick $k^\dagger = t\eta/\log(t\eta)^\beta$, which leads to $\sum_{i>k^\dagger} = \log(t\eta)^{1-\beta}$. Then Corollary 4.4 implies that

$$\mathbb{E}_{\mathrm{SGD},\mathbf{w}^*,\boldsymbol{\epsilon}}\big[\mathrm{FlutuationError}(\mathbf{w}_t)\big]$$

$$\lesssim \frac{\eta}{\log(1/\eta)} \cdot \left[\log(t)\log(n) + \frac{\log^{5/2}(n)t\eta}{n^{1/2}} \cdot \log(t\eta)^{1-\beta}\right] \cdot \min\left\{\frac{n\omega^2 + \sigma^2 \, \mathrm{tr}(\mathbf{A}^{-1})}{t\eta}, \omega^2 \, \mathrm{tr}(\mathbf{H}) + \sigma^2\right\}.$$

It is clear that when $t \to \infty$, we have

$$\mathbb{E}_{\mathrm{SGD},\mathbf{w}^*,\boldsymbol{\epsilon}}\big[\mathrm{FlutuationError}(\mathbf{w}_t)\big] \lesssim \frac{\eta(n\omega^2 + \sigma^2 \, \mathrm{tr}(\mathbf{A}^{-1}))}{\log(1/\eta)} \cdot \left[\frac{\log(t)}{t\eta}\log(n) + \frac{\log^{5/2}(n)}{n^{1/2}} \cdot \log(t\eta)^{1-\beta}\right]$$

$$\to 0. \tag{D.1}$$

Then we will move on to the GD error in Corollary 4.4, we can get that

$$\lambda_{k^*} \asymp \frac{1}{\eta t} + \frac{1}{n} \sum_{i > k^*} \lambda_i.$$

Plugging the fact that $\lambda_i = i^{-1} \log(i+1)^{-\beta}$, the above equality implies that

$$(k^*)^{-1} \log(k^*)^{-\beta} \asymp \frac{1}{\eta t} + \frac{1}{n} \log(k^*)^{1-\beta},$$

which further leads to

$$k^* = \min \left\{ \frac{t\eta}{\log(t\eta)^\beta}, \frac{n}{\log(n)} \right\}.$$

Note that when characterizing the upper bound of SGD, we can pick $k^*$ arbitrarily. Therefore, we will consider two cases accordingly: (1) $t\eta/\log(t\eta)^\beta \le n/\log(n)$; and (2) $t\eta/\log(t\eta)^\beta > n/\log(n)$. For the first case, we will pick $k^* = t\eta/\log(t\eta)^\beta$ and get that

$$\widetilde{\lambda} \asymp \frac{n}{t\eta} + \log(t\eta)^{1-\beta}.$$

Then given the value of $k^*$, we can further obtain that

$$\sum_{i \le k^*} \frac{1}{\lambda_i} = \sum_{i \le k^*} i \log(i+1)^\beta \asymp (k^*)^2 \log(k^*)^\beta \asymp \frac{(t\eta)^2}{\log(t\eta)^\beta}$$

$$\sum_{i \le k^*} \lambda_i \asymp \log(t\eta)^{1-\beta}$$

$$\sum_{i > k^*} \lambda_i^2 = \sum_{i > k^*} \frac{1}{i^2 \log(i+1)^{2\beta}} \asymp \frac{1}{k^* \log(k^*)^{2\beta}} \asymp \frac{1}{t\eta \log(t\eta)^\beta}.$$

Therefore, we can get that

$$\frac{\widetilde{\lambda}^2}{n^2} \cdot \sum_{i \le k^*} \frac{1}{\lambda_i} + \sum_{i > k^*} \lambda_i \asymp \log(t\eta)^{1-\beta}, \quad \frac{k^*}{n} + \frac{n}{\widetilde{\lambda}^2} \sum_{i > k^*} \lambda_i^2 \lesssim \frac{t\eta}{n \log(t\eta)^\beta}.$$

Then we can get the following according to Corollary 4.4,

$$\mathbb{E}_{\text{SGD}, \mathbf{w}^*, \boldsymbol{\epsilon}} \big[ \mathcal{E}(\mathbf{w}_t) \big] \lesssim \omega^2 \cdot \log(t\eta)^{1-\beta} + \sigma^2 \cdot \frac{t\eta}{n \log(t\eta)^\beta}$$

$$+ \frac{\eta}{\log(1/\eta)} \cdot \left[ \log(t) \log(n) + \frac{\log^{5/2}(n) t\eta}{n^{1/2}} \cdot \log(t\eta)^{1-\beta} \right] \cdot \min \left\{ \frac{n\omega^2 + \sigma^2 \operatorname{tr}(\mathbf{A}^{-1})}{t\eta}, \omega^2 \operatorname{tr}(\mathbf{H}) + \sigma^2 \right\},$$

Taking $t\eta = n$ and set

$$\eta \lesssim \log^{-3}(n) \cdot n^{-1/2},$$

we can immediately get that

$$\mathbb{E}_{\text{SGD}, \mathbf{w}^*, \boldsymbol{\epsilon}} [\mathcal{E}(\mathbf{w}_t)] \lesssim \omega^2 \cdot \log(n)^{1-\beta} + \sigma^2 \log(n)^{-\beta}.$$

For the second case of $t\eta/\log(t\eta)^\beta > n/\log(n)$, we will pick $k^* = n/\log(n)$, which leads to $\widetilde{\lambda} = \log(n)^{1-\beta}$. Then we can get

$$\sum_{i \le k^*} \frac{1}{\lambda_i} \asymp \frac{n^2}{\log(n)^{2-\beta}}, \quad \sum_{i \le k^*} \lambda_i \asymp \log(n)^{1-\beta}, \quad \sum_{i > k^*} \lambda_i^2 \asymp \frac{1}{n \log(n)^{2\beta-1}}.$$

This further leads to

$$\frac{\widetilde{\lambda}^2}{n^2} \cdot \sum_{i \le k^*} \frac{1}{\lambda_i} + \sum_{i > k^*} \lambda_i \asymp \log(n)^{1-\beta}, \quad \frac{k^*}{n} + \frac{n}{\widetilde{\lambda}^2} \sum_{i > k^*} \lambda_i^2 \asymp \frac{1}{\log(n)}.$$

Then we can get the following upper bound on the excess risk of SGD:

$$\mathbb{E}_{\text{SGD},\mathbf{w}^*,\boldsymbol{\epsilon}}\big[\mathcal{E}(\mathbf{w}_t)\big] \lesssim \omega^2 \cdot \log(n)^{1-\beta} + \sigma^2 \cdot \frac{1}{\log(n)}$$

$$+ \frac{\eta}{\log(1/\eta)} \cdot \left[\log(t)\log(n) + \frac{\log^{5/2}(n)t\eta}{n^{1/2}} \cdot \log(t\eta)^{1-\beta}\right] \cdot \min\left\{\frac{n\omega^2 + \sigma^2 \operatorname{tr}(\mathbf{A}^{-1})}{t\eta}, \omega^2 \operatorname{tr}(\mathbf{H}) + \sigma^2\right\}.$$

Taking $t \to \infty$ and applying (D.1) give

$$\lim_{t\to\infty} \mathbb{E}_{\text{SGD},\mathbf{w}^*,\boldsymbol{\epsilon}}[\mathcal{E}(\mathbf{w}_t)] \lesssim \omega^2 \cdot \log(n)^{1-\beta} + \sigma^2 \log(n)^{-1}.$$

This completes the proof.

∎

## D.3 Proof of Corollary 4.6

**Proof.** [Proof of Corollary 4.6] We will first calculate $k^*$ defined in Corollary 4.4. Note that

$$k^* = \min\left\{k : n\lambda_{k+1} \leq \frac{n}{\eta t} + \sum_{i>k}\lambda_i\right\},$$

and $\sum_{i>k}\lambda_i = \sum_{i>k}i^{-1-r} \asymp k^{-r}$. Then, it can be shown that

$$k^* = (t\eta)^{1/(r+1)}. \tag{D.2}$$

Recall that Corollary 4.4 shows

$$\mathbb{E}_{\text{SGD},\mathbf{w}^*}[\text{Risk}(\mathbf{w}_t)]$$

$$\lesssim \omega^2 \cdot \underbrace{\left(\frac{\widetilde{\lambda}^2}{n^2} \cdot \sum_{i\leq k^*}\frac{1}{\lambda_i} + \sum_{i>k^*}\lambda_i\right)}_{I_1} + \sigma^2 \cdot \underbrace{\left(\frac{k^*}{n} + \frac{n}{\widetilde{\lambda}^2}\sum_{i>k^*}\lambda_i^2\right)}_{I_2} + (\omega^2\operatorname{tr}(\mathbf{H}) + \sigma^2))\eta$$

$$\cdot \underbrace{\left[\log(t) \cdot \left(\operatorname{tr}(\mathbf{H})\log(n) + \frac{k^*\log^{5/2}(n)}{n^{1/2}}\right) + \frac{\log^{5/2}(n)t\eta}{n^{1/2}} \cdot \sum_{i>k^*}\lambda_i\right)\right]}_{I_3}.$$

Then, applying (D.2) gives

$$\sum_{i>k^*}\lambda_i \asymp (k^*)^{-r} \asymp (t\eta)^{-r/(r+1)}, \quad \sum_{i>k^*}\lambda_i^2 \asymp (k^*)^{-2r-1} \asymp (t\eta)^{-(2r+1)/(r+1)},$$

$$\sum_{i\leq k^*}\frac{1}{\lambda_i} \asymp (k^*)^{r+2} = (t\eta)^{(r+2)/(r+1)}, \quad \widetilde{\lambda} \asymp \frac{n}{t\eta}, \quad \operatorname{tr}(\mathbf{H}) \asymp 1$$

Putting the above into the formula of $I_1$, $I_2$, and $I_3$, we can get

$$I_1 \lesssim \frac{n^2/(t\eta)^2}{n^2} \cdot (t\eta)^{(r+2)/(r+1)} + (t\eta)^{-r/(r+1)} \asymp (t\eta)^{-r/(r+1)};$$

$$I_2 \lesssim \frac{(t\eta)^{1/(r+1)}}{n} + \frac{n}{n^2/(t\eta)^2} \cdot (t\eta)^{-(2r+1)/(r+1)} \asymp \frac{(t\eta)^{1/(r+1)}}{n};$$

$$I_3 \lesssim \log(t) \cdot \left(\log(n) + \frac{(t\eta)^{1/(r+1)}\log^{5/2}(n)}{n^{1/2}}\right) + \frac{\log^{5/2}(n)t\eta}{n^{1/2}} \cdot (t\eta)^{-r/(r+1)}\right)$$

$$\lesssim \log(t) \cdot \left[\log(n) + \frac{\log^{5/2}(n)}{n^{1/2}} \cdot (t\eta)^{1/(r+Corollary1)}\right].$$

Combining the above results leads to

$$\mathbb{E}_{\text{SGD},\mathbf{w}^*}[\text{Risk}(\mathbf{w}_t)] \lesssim \omega^2 \cdot (t\eta)^{-r/(r+1)} + \sigma^2 \cdot \frac{(t\eta)^{1/(r+1)}}{n}$$

$$+ (\omega^2 + \sigma^2) \cdot \eta \cdot \log(t) \cdot \left[ \log(n) + \frac{\log^{5/2}(n)}{n^{1/2}} \cdot (t\eta)^{1/(r+1)} \right].$$

This completes the proof.

∎

$$+ (\omega^2 + \sigma^2) \cdot \eta \cdot \log(t) \cdot \left[ \log(n) + \frac{\log^{5/2}(n)}{n^{1/2}} \cdot (t\eta)^{1/(r+1)} \right].$$