# OpenReview forum: "Risk Bounds of Multi-Pass SGD for Least Squares  in  the Interpolation Regime"
_NeurIPS.cc/2022/Conference — NeurIPS 2022 Accept_

### Official Review · Reviewer_12me · 2022-07-08

**Rating:** 8
**Confidence:** 5
**Soundness:** 4 excellent
**Presentation:** 3 good
**Contribution:** 4 excellent

**Summary:**

This paper studies the generalization properties of over-parameterized least squares with SGD. To be specific, the considered SGD works in a setting of multi-pass, one batch, last iterate, constant step-size SGD with replacement. By considering SGD with replacement, the analysis can be simplified due to the unbiased property of stochastic gradients, and accordingly, the iid data sampling strategy still holds under the multi-pass setting.


**Questions:**

1) Line 120: The expected covariance matrix H is required to be positive definite and A is full rank?

2) Line 140: Running GD under the interpolation regime converges to the min-norm interpolator. This is a well-known result. However, the convergence of SGD path depends on the choice of the ground truth. It’s unclear to me on the convergence of SGD under the interpolation regime.

3) Line 162: theorem 4.1, I like this theorem as well as (5.3) to show that, the excess risk of SGD is always worse than GD under the same iterations. I’m wondering that, this still holds true for other problem settings? e.g., different loss functions, over-parameterized neural networks. Besides, this result needs to be well discussed when compared to results on implicit bias of SGD.

4) What’s the result under the exponential decay? As the authors mentioned in line 236, SGD with fixed size can only hold for small step-size under the polynomial decay. It seems that Corollary 4.5 allows for large step-size?

5) albeit iid, the data are sampled from the empirical measure instead of the true (but unknown) population distribution \mathcal{D}. Although the empirical measure converges to such true measure a.s. by the strong law of large numbers, it would be better to clarify this as the convergence rate of them needs to be estimated.

6) Line 610: $\Sigma$ is a random matrix, how does the choice of step-size can ensure $I – \eta \Sigma$ is PSD as $\eta$ is in a deterministic upper bound.

Minor issues:

1) some notations are relatively confusing, e.g., $\bf \Theta_1$, $\Theta_1$, and $\Theta$.

2) Line197: Lemma 5.2 requires the assumption $\lambda_1 = \Theta(1)$, please specify it.

3) line 540: $nr$ -> $nr^2$ follows $\chi^2(n)$ distribution. And why $r \geq 1/2$ is satisfied?

Typo:

Line 272: $\hat{\bf w}_t$ is missing in ${\bf \Theta}_1$

Line 675 and line 676 are repeated.


**Limitations:**

yes

**Strengths And Weaknesses:**

Pros:

This paper provides a solid analysis of multi-pass SGD of least squares in the interpolation regime. It provides nice results, e.g., 1) under this setting, the excess risk of SGD is always worse than that of GD; 2) the fluctuation error bound as well as the excess risk is based the eigenvalue decay of the data; 3) at the same level of excess risk, SGD requires more iterations but less gradient complexity than GD.

Cons:

I think this work has no significant drawback, but it would better to clarify the theoretical results in several aspects:

1) Line 261: what’s the difference between the expectation over “SGD” and $i_t$? According to the randomness definition of “SGD” in line 155, the randomness only comes from uniformly at random, i.e., “SGD”$=(i_1, i_2, …, i_t, …)$, right? This needs to be clear, especially on $\mathbb{E}_{i_t} {x}_{i_t} {x}_{i_t}^{T}= {\Sigma}$.

2) What’s the benefit of multi-pass version as I don’t find very clear benefit in Corollary 4.5 under a relatively slow eigenvalue decay.

---

> ### Author Response · Authors · 2022-08-02
> **Response to Reviewer 12me, Part 1/2**
>
> Thanks for your positive comments.
>
> ---
>
> **Q1**: "what’s the difference between the expectation over “SGD” and $i_t$? According to the randomness definition of “SGD” in line 155, the randomness only comes from uniformly at random, i.e., “SGD” $=(i_1,i_2,…,i_t,…)$,, right? This needs to be clear, especially on $\mathbb{E}{i_t} {x}{i_t} {x}_{i_t}^{T}= {\Sigma}$."
>
> **A1**: You are correct, the expectation over SGD is the expectation of all randomly drawn data indices $(i_1, \dots,i_t,\dots,)$. The expectation over $i_t$ is the expectation over the randomly sampled data at iteration $t$. Throughout our theoretical analysis, we repeatedly use $E_{SGD}[f(w_{t+1})] = E_{i_1, \dots,i_t}[f(w_{t+1})] = E_{i_1,\dots,t_{t-1}}[E_{i_t|i_1,\dots,t_{t-1}}[f(w_{t+1})]]$. When calculating the inner expectation, we may encounter something like $E_{i_t|i_1,\dots,t_{t-1}}[x_{i_t}x_{i_t}^\top]$. Then using the fact that $i_t$ is independent of $i_1,\dots,t_{t-1}$, we have $E_{i_t|i_1,\dots,t_{t-1}}[x_{i_t}x_{i_t}^\top] = E_{i_t}[x_{i_t}x_{i_t}^\top]=\Sigma$. We have clarified this in the revision.
>
> ---
>
> **Q2**: "What’s the benefit of multi-pass version as I don’t find very clear benefit in Corollary 4.5 under a relatively slow eigenvalue decay"
>
> **A2**: Corollary 4.5 actually compares the generalization bounds of multi-pass SGD and minimum-norm solution. It is clear that SGD always outperforms the minimum-norm solution when $\beta>2$, i..e,  the data covariance has a decay that is faster than $i^{-2}$. This is because that the SGD solution achieves a $\tilde O(\log(n)^{1-\beta})$ risk bound while the minimum-norm solution can only achieve a $\tilde O(\log(n)^{-1})$ risk bound.
>
> ---
>
> **Q3**: "The expected covariance matrix H is required to be positive definite and A is full rank?"
>
> **A3**: Yes, we require H to be positive definite so that A is full rank almost surely. We have added this assumption in the revision.
>
> ---
>
> **Q4**: "Line 140: Running GD under the interpolation regime converges to the min-norm interpolator. This is a well-known result. However, the convergence of SGD path depends on the choice of the ground truth. It’s unclear to me on the convergence of SGD under the interpolation regime."
>
> **A4**: Based on our Theorems 4.1 and 4.2, the convergence of SGD to the minimum-norm interpolator can be characterized by showing the fluctuation error goes to zero, which can be guaranteed if the minimum-norm interpolator has a bounded $\ell_2$ norm.
>
> ---
>
> **Q5**: theorem 4.1, I like this theorem as well as (5.3) to show that, the excess risk of SGD is always worse than GD under the same iterations. I’m wondering that, this still holds true for other problem settings? e.g., different loss functions, over-parameterized neural networks. Besides, this result needs to be well discussed when compared to results on implicit bias of SGD.
>
> **A5**: The technical analysis is largely built based on the least square problem. Therefore, we suspect that the analysis may not be simply extended to other loss functions (i.e., cross-entropy loss). Besides, we would like to point out that certain over-parameterized neural networks can be covered by our theory. Particularly, when the neural network width is sufficiently large or even infinite, the neural network function can be interpreted as a linear model on the random feature (or kernel model). Then our theory is possible to be applied to show the worse generalization ability of SGD compared to GD in this setting.
>
> Regarding the implicit bias of SGD, prior works have shown that given an infinite number of iterations, SGD and GD will both converge to the minimum-norm interpolator for least square problems. Our theory also suggests this since the fluctuation error will become zero as $T$ goes to infinity.

---

> > ### Comment · Reviewer_12me · 2022-08-05
> > **more discussion on implicit bias of SGD**
> >
> > Thanks for the author's response.
> >
> > I suggest the authors give a relatively detail discussion on implicit bias of SGD that helps generalization in the final version.
> > One hand, SGD has larger excess risk than GD, but on the other hand, under some slow data spectrum decay case e.g., Corollary 4.5, SGD always outperforms the minimum-norm solution.
> >
> > I think clarification on this point will increase the significance of this work.
> >
> > Besides, if a lower bound can be also provided, this will be more convince, but this would be beyond the scope of current version.

---

> > > ### Author Response · Authors · 2022-08-06
> > > **Thank you for your suggestion**
> > >
> > > Thank you very much for your suggestion.
> > >
> > > For linear regression problems, when the iteration number goes to infinity, both SGD (with a sufficiently small learning rate) and GD will converge to the minimum-norm solution. This is the implicit bias of SGD/GD when $T\rightarrow \infty$. However, in practice, one often does early stopping, which stops SGD/GD after a finite number of iterations. Within a finite number of iterations, the performance of SGD and GD can be different.
> > >
> > > In order to study SGD and compare it with minimum-norm solution, a complete characterization of the implicit bias of SGD should not only focus on the limiting point, but also take other factors into consideration (e.g., finite iteration number $T$, large learning rate $\eta$, etc). This is partly achieved by Corollary 4.4, which characterizes the risk of SGD after a finite number of iterations with a certain learning rate, rather than its limiting point.
> > >
> > > Furthermore, one can tune the iteration number $T$ (i.e., early stopping) and learning rate $\eta$ to get a better generalization. This is demonstrated by our Corollary 4.5 where the iteration number and learning rate of SGD are properly tuned, so that SGD can outperform the minimum-norm solution (due to early stopping). Besides, Theorem 4.1 shows that SGD is worse than GD when using the same iteration number and learning rate, this also suggests that GD, with proper early stopping, can also outperform the minimum-norm solution. We will add more discussion on this point in the final version (given the additional page).

---

> ### Author Response · Authors · 2022-08-02
> **Response to Reviewer 12me, Part 2/2**
>
> **Q6**: What’s the result under the exponential decay? As the authors mentioned in line 236, SGD with fixed size can only hold for small step-size under the polynomial decay. It seems that Corollary 4.5 allows for large step-size?
>
> **A6**: We will add the convergence result under the exponential decay of the eigenspectrum in the revision, which is nearly equivalent to taking the limit $r\rightarrow \infty$ in the polynomial decay results. Besides, we would like to clarify that in Corollaries 4.6 and line 236, the step size is chosen to achieve the desired excess risk bound, this does not mean that the stepsize chosen in line 236 needs to be that small to guarantee generalization. In fact, our general excess risk bound (Corollary 4.4) holds for arbitrary $\eta \lesssim 1/\mathrm{tr}(H)$ and $\eta = o(1/\log(t)\log(n))$ is sufficient to guarantee diminishing generalization error when $\omega^2, \sigma^2 = O(1)$.
>
> ---
>
> **Q7**: albeit iid, the data are sampled from the empirical measure instead of the true (but unknown) population distribution $\mathcal{D}$. Although the empirical measure converges to such true measure a.s. by the strong law of large numbers, it would be better to clarify this as the convergence rate of them needs to be estimated.
>
> **A7**: It is true that the empirical covariance $\Sigma$ will converge to the population covariance $H$ as $n$ goes to infinity. The estimation error of $\Sigma$ has already been considered in our theoretical analysis  (See Lemma B.2 and its usage in Line 588-590).
>
> ---
>
> **Q8**: $\Sigma$ is a random matrix, how does the choice of step-size can ensure $I - \eta \Sigma$ is PSD as $\eta$ is in a deterministic upper bound.
>
> **A8**: In fact, by concentration result (e.g., Lemma B.2 in our paper), we can prove that with sufficiently high probability, the largest eigenvalue of $\Sigma$ is upper bounded by a constant. Then, with high probability (e.g., $1-1/\mathrm{poly}(n)$), we can guarantee that $I - \eta \Sigma$ is PSD. This high probability argument is also stated in all Theorems and Corollaries in our paper.
>
> ---
>
> **Q9**: Minor issues:
>
> **A9**: We have fixed them in the revision.

---

### Official Review · Reviewer_BSbS · 2022-07-09

**Rating:** 6
**Confidence:** 3
**Soundness:** 3 good
**Presentation:** 4 excellent
**Contribution:** 3 good

**Summary:**

This work studies the statistical aspects of empirical risk minimization (ERM) when solving it using SGD. The authors particularly focus on the (un-regularized) least squares objective where the minimum norm solution to the ERM problem can interpolate the empirical data. They show that the excess risk of SGD is exactly that of GD plus a non-negative fluctuation error under the same hyperparameters. This decomposition is used to give problem-dependent risk bounds for SGD and GD. These bounds show that GD achieves lower excess risk than SGD when given the same number of iterations, but SGD has a lower iteration cost so it can still be preferred in terms of computation.

**Questions:**

minor issues:

- "gram" matrix should be spelled as "Gram" matrix
- When comparing with existing results (page 6), it might be helpful to briefly introduce what a *source condition* is, for those who are less familiar with the related work.

**Limitations:**

I believe the authors have adequately addressed the limitations, such as lacking results for diminishing step sizes and the multi-pass SGD in the without replacement sampling regime. I agree with the authors on the foreseeable difficulties with these analyses and that they should be considered interesting future directions.

**Strengths And Weaknesses:**

This paper is well organized and highly polished. The claims are mostly sound, although I have only checked the proofs at a high level. I find the risk decomposition result interesting, and that giving problem-dependent risk bound is indeed desired to reflect the inherent difficulty of individual problems. Another strength is that the assumptions made in this paper are more general than existing works analyzing similar problems, although I have one issue regarding the combination of the assumptions together:

Although the authors conjecture that the prior assumption on $w^*$ can potentially be removed, the results still rely on that to be true. However, it is unclear to me why, when a prior is assumed but the updates themselves are solving for an MLE rather than a MAP estimation. That is, if we know that the ground truth comes from Normal distribution with a known covariance that is a multiple of the diagonal, then the most natural thing to do for statistical accuracy is to incorporate that information by imposing a L2 regularization proportional to that multiple. However, if we do that, it'll be uncertain whether the interpolation condition can still easily hold. The minimum norm solution should also have an additional diagonal added to $X^TX$. To me this is a strange setting where a prior is assumed but never used in the algorithm, and I'm not sure how to interpret that, or whether it can be incorporated into the algorithm without contradicting the interpolation assumption. It would be great if the authors could elaborate on the reasoning behind throwing away the prior information, if I understood it correctly.

Lastly, although the authors have fully acknowledged that their results apply only to vanilla SGD in the with replacement setting, and the without replacement setting is set for future work, I still find the "multi-pass SGD" terminology somewhat misleading. In with replacement SGD a single pass does not guarantee a full pass of the training data - that is only guaranteed by some permutation-based SGD. This is not intended to nitpick (and was not a scoring factor in my evaluation) but I would like to encourage the authors to find an alternative expression.

---

> ### Author Response · Authors · 2022-08-02
> **Response to Reviewer BSbS**
>
> Thanks for your valuable and constructive comments.
>
> ---
> **Q1**: Why is the prior assumed but not used?
>
> **A1**:  We would like to emphasize that the prior assumption is made for the purpose of technical analysis, not algorithm design (e.g., L2 regularization). In detail, in the proof of Theorem 4.3 (GD risk), we need the prior distribution to make the analysis go through. On the other hand, the equality-based decomposition of the SGD excess risk relies on the interpolation property. Like you said, if we do MAP (L2 regularization), then the interpolation property of $\hat{w}$ will no longer hold, and this will also affect the risk bounds of SGD in our paper (e.g., Corollary 4.4). We agree that it is an interesting question to study multipass SGD on a regularized objective (e.g., L2 regularization) and we will study it in the future.
>
> ---
>
> **Q2**: Usage of "multi-pass SGD"
>
> **A2**: Thank you for your advice. We recognize this subtlety here. The reason we use multi-pass SGD is mainly that we want to follow the terminology used in several prior works [Lin and Rosasco 2017, Pillaud-Vivien et al. 2018, Mücke et al. 2019].
>
> ---
>
> **Q3**: Minor issues
>
> **A3**: We have fixed them in the revision.

---

### Official Review · Reviewer_t7KM · 2022-07-11

**Rating:** 6
**Confidence:** 4
**Soundness:** 3 good
**Presentation:** 3 good
**Contribution:** 3 good

**Summary:**

This work studies the generalization capability of multi-pass SGD for least squares problems under the interpolation regime. Specifically, the excess risk bound using the spectrum of data covariance achieved is provided under the assumptions: Gaussian prior to the true parameter and existence of interpolator. This result demonstrates the advantage of multi-pass SGD over OLS and GD under specific spectrum settings.


**Questions:**

- On lines 170-173: I am not sure if Young’s inequality (appearing in related works) is a weak point which should be noted. I think it does not slow down the convergence rate. What are the advantages of not using it?

- What happens when taking limit $d\to \infty$ in proportion to $n$ as done in the context of random matrix theory? This question relates to a weakness stated (1) above. I am wondering if zero excess risk is achieved by taking limits of $n, d$, simultaneously.






**Strengths And Weaknesses:**

**Strengths**

A contribution of this work is that the excess risk bound described with the spectrum of data covariance is derived under the existence of an interpolator for examples, which seems to be new in the context of multi-pass SGD analysis (Pillaud-Vivien et al. [27], Line and Rosasco [21], and Mucke et al. [24]). Indeed, these studies assume specific decay rates of the spectrum and source conditions on the true parameter.


Although direct comparison with these existing results is difficult in general, the authors show that multi-pass SGD achieves at least the comparable excess risk bound with OLS under a specific spectrum setting (Corollary 4.5) as a sanity check.


**Weaknesses**

(1) The existence of an interpolator limits the number of examples $n$ to e.g., $n<d$ (where $d$ is dimensionality.) This means we cannot take the limit $n\to \infty$ and cannot show the convergence to zero excess risk. I think this limitation seems unavoidable because the convergence $\hat{w}\to w_*$ (under $n\to\infty$) contradicts the assumptions where $\hat{w}$ has finite norm and $w_*$ follows Gaussian prior.

(2) On one hand, the assumption of the Gaussian prior is nice because it is a milder assumption compared to the source condition. On the other hand, this hinders characterizing learnable classes of target functions (e.g., Sobolev class et al.).


**Other comments**

- On line 86-87: ``More recently, ... an exponential rate in the interpolation regime [22,5,33,34]"
An exponential convergence of SGD under the interpolation regime was also shown in Needell et al. [25].  Note that $\sigma^2=0$ (in Theorem 2.1, [25]) which is attained in the interpolation regime yields an exponential rate.

---

> ### Author Response · Authors · 2022-08-02
> **Response to Reviewer t7KM**
>
>
> Thanks for recognizing the strengths of our paper.
>
> ---
> **Q1**:  “The existence of an interpolator limits the number of examples $n$ to e.g., $n<d$, This means we cannot take the limit $n\rightarrow \infty$, and cannot show the convergence to zero excess risk. I think this limitation seems unavoidable because the convergence $\hat w \rightarrow w^*$ contradicts the assumptions”
>
> **A1**: We agree that our results cannot be applied to the setting where $d < n$ due to the requirement of the existence of an interpolator. However, this does not prevent from getting a diminishing excess risk when taking the limit $n\rightarrow \infty$, because we can also let $d$ go to $\infty$. In particular, in the infinite-dimensional case where the eigenvalues of the data covariance have a relatively fast decaying, as shown in Corollary 4.5 and the discussion after Corollary 4.6, both GD and SGD, with proper learning rates, can achieve $\tilde O(1/\mathrm{poly}(n))$ or $\tilde O(1/\mathrm{polylog}(n))$ excess risk bounds.
>
> ---
>
> **Q2**: "On line 86-87: ``More recently, ... an exponential rate in the interpolation regime [22,5,33,34]" An exponential convergence of SGD under the interpolation regime was also shown in Needell et al. [25]. Note that $\sigma^2=0$ (in Theorem 2.1, [25]) which is attained in the interpolation regime yields an exponential rate."
>
> **A2**: Thanks for pointing that out. We have added the reference [25] when stating the exponential rate of SGD in the interpolation regime.
>
> ---
>
> **Q3**: "I am not sure if Young’s inequality (appearing in related works) is a weak point which should be noted. I think it does not slow down the convergence rate. What are the advantages of not using it?"
>
> **A3**: It is true that using Young’s inequality does not significantly affect the convergence rate (only involving constant factors) and one can definitely use Young’s inequality for establishing the upper bound of the convergence for SGD.  However, Young’s inequality cannot be applied to demonstrate a worse generalization performance of multi-pass SGD compared to GD as it is stated as an “upper bound”. In our Theorem 4.1, the risk of SGD is stated as an equality, which can show that SGD has a higher generalization error than GD.
>
> ---
>
> **Q4**: "What happens when taking limit $d\rightarrow \infty$  in proportion to $n$ as done in the context of random matrix theory? I am wondering if zero excess risk is achieved by taking limits of $n, d$, simultaneously. "
>
> **A4**: We would like to clarify that our setting is different from the so-called “proportion limit” setting since we consider the non-asymptotic excess risk bounds of SGD, i.e., we aim to provide an explicit dependency on the finite training sample size $n$.
> To your second question, our theory can indeed imply zero excess risk when taking limits of $n, d$ simultaneously when the eigen spectrums have certain decay properties. In particular, in Corollary 4.5 and the discussion after Corollary 4.6, we show that as long as $d>n$ (to guarantee the existence of an interpolator), both GD and SGD with proper learning rates can achieve $\tilde O(1/\mathrm{polylog}(n))$ or $\tilde O(1/\mathrm{poly}(n))$ excess risk bounds when the eigenspectrum has a polynomial-log or polynomial decay. It can be immediately seen that the excess risk diminishes as $n,d$ goes to $\infty$.

---

> > ### Comment · Reviewer_t7KM · 2022-08-09
> > **Thanks for the response.**
> >
> > This additional explanation would be nice, which will improve the quality of the paper. I would like to keep the evaluation.

---

### Official Review · Reviewer_UZch · 2022-07-11

**Rating:** 6
**Confidence:** 5
**Soundness:** 3 good
**Presentation:** 3 good
**Contribution:** 2 fair

**Summary:**

The authors of the paper present their work on multipass SGD for least-squares. They show the following contributions:
- The excess risk of SGD is the one of GD plus a positive fluctuation term
- They show data-dependant bound consistent in the $t\to\infty$ regime, where $t$ is the number of iterations



**Questions:**

I have no question regarding the results.

**Limitations:**

- Hypothesis on the data to be subgaussian and indepedant.
- Very similar to previous work.

**Strengths And Weaknesses:**

### **Strengths**

- The authors present a clear analysis of the setting they describe: an excess risk upper bound for multipass SGD for Least-squares. As far as I understood, they do so for the *final iterate*, which has been rarely considered. I guess that the authors' could stress on this idea a little bit more as previous analysis do not cover this case.
- They provide bounds that are consistent in the so-called *benign overfitting regime*


### **Weaknesses**

- The main weakness is that the paper is largely overselling: e.g. the fact that the excess risk of SGD decomposes as GD and a fluctuation error cannot really be considered as a deep contribution even if previous work did not write it explicitly. In the same manner, comparison with existing literature is misleading: what the authors say about the novelty concerning the adaptivity to covariance structure is simply false.
- Indeed, the authors claim that one of the main difference of their bound is that they are data dependant whereas previous work are not. I truly think that there is a confusion with the capacity and source conditions used by the mentioned authors: they correspond exactly to a nice framework showing how adaptive to the data SGD/GD/Ridge is. What is strange is that the authors, to exemplify their bound, take specific polynomial or polynomial + log decrease of eigenvalue of the covariance matrix: that is exactly a *capacity model* of the data.  Going further, it is said that the authors need a Gaussian prior over $w_*$. That condition is indeed not needed in the previous work. Here, the *source condition* could be a deeper look on how $w_*$ has been chosen (think about a *Gaussian process*) and would simply extend the authors' work.
- The data assumptions on $H^{-1/2} x$ is strong (especially the independence of the coordinates). This almost corresponds to a gaussian model of the data. It may be necessary for the benign overfitting phenomenon to be exhibited but as far as the SGD optimisation is concerned, this is a very restrictive assumption compared to previous work.
- Finally, I have to say that I find quite curious that the authors show bounds in the main theorem that necessitate a deep look to be understood (see corollary 4.4 for example). I understand that this appears to be one of the claimed strength of the authors but it is simply misleading for me.

---

> ### Author Response · Authors · 2022-08-02
> **Response to Reviewer UZch**
>
> **Q1**:  "The fact that the excess risk of SGD decomposes as GD and a fluctuation error cannot really be considered as a deep contribution even if previous work did not write it explicitly. "
>
> **A1**: Indeed, as mentioned in our paper, existing works have adopted a similar idea for decomposing SGD error as GD error plus a fluctuation error. Yet, to the best of our knowledge, we are the first to provide a decomposition in equality instead of inequality. Without a decomposition in an equality form, one cannot rigorously justify that the SGD risk is no less than that of GD.
>
> ---
>
> **Q2**: "Comparison with existing literature is misleading. I truly think that there is a confusion with the capacity and source conditions used by the mentioned authors: they correspond exactly to a nice framework showing how adaptive to the data SGD/GD/Ridge is. what the authors say about the novelty concerning the adaptivity to covariance structure is simply false. What is strange is that the authors, to exemplify their bound, take specific polynomial or polynomial + log decrease of eigenvalue of the covariance matrix: that is exactly a capacity model of the data."
>
> **A2**: We agree with the reviewer that the result in [Pillaud-Vivien et al., 2018] is indeed adaptive to the data. However, their adaptivity is governed by a single parameter $\alpha$, which appears in the capacity assumption $\mathrm{tr}(H^{1/\alpha})\le \infty$. Therefore, if there are two covariance matrix $\mathbf{\Sigma}_1$ and $\mathbf{\Sigma_2}$ that satisfies the capacity assumption with the same $\alpha$, then their results cannot distinguish these two linear regression instances.  In contrast, our excess risk bound is established in an instance-wise manner, and its adaptivity is characterized as the entire eigenspectrum of the data covariance (i.e., a function of $\lambda_1,\lambda_2,\dots$). Therefore, our result is more fine-grained and the adaptivity of our result is strictly stronger than that in [Pillaud-Vivien et al., 2018]. We have rephrased the comparison with [Pillaud-Vivien et al., 2018] in the revised paper to avoid any confusion. Thank you for your suggestion.
>
> Regarding the polynomial+log decrease of eigenvalues of the covariance matrix, this is an example of the covariance matrix originally studied in [Bartlett et al., 2019] and later considered in [Zou et al., 2021]. For this instance (related to benign overfitting), capacity model-based analysis cannot give a diminishing excess risk bound to our knowledge.
>
> ---
>
> **Q3**: "it is said that the authors need a Gaussian prior over $w^*$. That condition is indeed not needed in the previous work. Here, the source condition could be a deeper look on how $w^*$ has been chosen (think about a Gaussian process) and would simply extend the authors' work."
>
> **A3**: Thanks for your suggestion, we agree that extending our results to the setting with source condition is a promising future direction. Yet we would like to point out that the Gaussian prior condition is not directly comparable with the source condition made in existing works. For instance, under the condition $\mathrm{tr}(H)< \infty$ (made in our paper), the Gaussian prior $N(0, \omega^2\mathbf{I})$ nearly implies that the source condition $\\|H\^{1/2-r}w\^*\\|\_2< \infty$ (Assumption A5 in [Pillaud-Vivien et al., 2018]) holds with $r=0$. Then Theorem 1 in [Pillaud-Vivien et. al., 2018] cannot be applied to prove the convergence of multi-pass SGD in this setting since its excess risk bound becomes a constant.
>
> ---
>
> **Q4**: "The data assumptions on $H^{-1/2}x$ is strong (especially the independence of the coordinates). This almost corresponds to a gaussian model of the data. It may be necessary for the benign overfitting phenomenon to be exhibited but as far as the SGD optimisation is concerned, this is a very restrictive assumption compared to previous work."
>
> **A4**: We agree that the data assumptions on $H^{-1/2}x$  are not standard in the prior SGD optimization works. However, they are commonly made in the analysis of high-dimensional least square problems [Dobriban and Wager, 2018]. Besides, this assumption is needed since we want to cover the benign overfitting regime [Bartlett et al. 2019]. We have added a comment in the revised paper to emphasize this limitation.

---

> > ### Comment · Reviewer_UZch · 2022-08-07
> > **Rebutall**
> >
> > **A1**: Alright.
> >
> > **A2**: Bounding w.r.t power of traces make only differences between polynomial decrease but finer rates can be considered. These "power law" capacity conditions are just here for the sake of clarity of bounds. I do not consider the author's bound sas more adaptive because they let the sum of eigenvalues explicitly in the Theorems.
> >
> > **A3**: No, the Gaussian prior $N(0, w^2 I)$ implies that the source condition "almost" corresponds to $r = 0$ (as stated in Assumption A5 in [Pillaud-Vivien et al., 2018]). Indeed, this is "almost" a specified case as, for some $C>0$, the norm $ | w^*|^2_2 < C w^2$  with high-probability. I agree that strictly speaking this assumption is not comparable, but with high probability over $w^*$, this is the case that $r = 0$. The authors should absolutely change this as, for now, there is no improvement in this direction. This also mislead other reviewers that wrote that this assumption is milder than a source condition.
> >
> > **A4**: Alright.

---

> > > ### Author Response · Authors · 2022-08-08
> > > **Re: Rebuttal**
> > >
> > > **Followup on Q2**: Bounding w.r.t power of traces make only differences between polynomial decrease but finer rates can be considered. These "power law" capacity conditions are just here for the sake of clarity of bounds. I do not consider the author's bound sas more adaptive because they let the sum of eigenvalues explicitly in the Theorems.
> > >
> > > **A**: If the reviewer still thinks our bound is not more *adaptive* than the source condition based bounds, we will withdraw this claim, because it looks like we have different understandings of “adaptivity”. We will just claim that our bound is an instance-wise bound that holds for each linear regression problem associated with a particular data covariance matrix. As a comparison, the bounds based on the source condition are the same for a class of linear regression problems whose covariance matrices satisfy the source condition parameterized by $\alpha$. So their bound is not an instance-wise bound.
> > >
> > > ---
> > > **Followup on Q3:** No, the Gaussian prior $N(0, \omega^2 I)$ implies that the source condition "almost" corresponds to $r=0$
> > >  (as stated in Assumption A5 in [Pillaud-Vivien et al., 2018]). Indeed, this is "almost" a specified case as, for some $C>0$, $\\|w\^\*\\|\_2\^2<C\\omega\^2$ with high-probability. I agree that strictly speaking this assumption is not comparable, but with high probability over $w^*$, this is the case that $r=0$. The authors should absolutely change this as, for now, there is no improvement in this direction. This also mislead other reviewers that wrote that this assumption is milder than a source condition.
> > >
> > > **A**: We think there is a misunderstanding by the reviewer. Our reasoning is correct. Let us first write down the condition on $w\^\*$:
> > > $$
> > > \\|H\^{1/2-r} w\^\*\\|_2^2< \\infty.
> > > $$
> > > Note that the exponent in the above condition is $1/2-r$, not $r$. So $N(0, \\omega\^2 I)$ corresponds to the source condition with $r=0$. In fact, we can only show that $\\|H\^{1/2}w\^\*\\|\_2\^2<C\\cdot \\omega\^2\\cdot \\mathrm{tr}(H)$ with high probability, which corresponds to the source condition $\\|H\^{1/2-r} w\^\*\\|\_2\^2< \\infty$ with $r=0$. Additionally, for the Gaussian prior, we have  $\\|w^*\\|\_2\^2<C\\cdot d\\cdot \\omega\^2$ rather than  $\\|w\^\*\\|\_2\^2<C\ \omega\^2$ with high probability (there is a factor of $d$ difference). So the bound will explode when $d\\rightarrow \\infty$. We guess your reasoning might have been based on $r=1/2$.
> > >
> > > In the original submission, we have said “Moreover, we assume $w\^\*$ follows a Gaussian prior (Assumption 3.1C), which is also not directly comparable to the source condition in existing works.” We don’t think there is anything misleading there, but we will make this point even clearer in the revision according to your suggestion.
> > >
> > > Please let us know if you have any further questions. Thank you.

---

> > > > ### Comment · Reviewer_UZch · 2022-08-08
> > > > **After second answer by authors**
> > > >
> > > > **Q2**: I thank the authors for their answer, they should do what they think is preferable. I tend to think that this is a bit overselling to claim for better adaptivity, but this is only a matter of taste.
> > > >
> > > > **Q3**: I am truly sorry for my previous remark and acknowledge that the authors are totally right on that subject: their setup corresponds indeed to $r=0$. I was very puzzled as in fact, it is *impossible* (in the minimax sense) to show a *polynomial* rate of convergence when $r = 0$ [see e.g. Steinwart results]. Instead a *logarithmic convergence rate* is shown. If the authors want to dig into the equivalence between random priors and capacity conditions, I recommend to have a look on the nice survey [https://arxiv.org/abs/1807.02582, Sections 3-4]. Once again, sorry for my mistake.
> > > >
> > > > I'll raise my score in accordance to my new understanding: but cannot raise it above $6$ as I do not feel that the results carries some significant novelty (both technically or for the overall benign overfitting/SGD understanding).

---

> > > > > ### Author Response · Authors · 2022-08-08
> > > > > **Thank you!**
> > > > >
> > > > > Thank you for pointing out the survey paper and for raising the score. We will read it and try to figure out the deeper relationship between random priors and the capacity condition.

---

### Meta-Review · Area_Chair_HVQn · 2022-08-21

**Recommendation:** Accept
**Confidence:** Certain

**Metareview:**

The reviewers have all found that the paper has many interesting ideas and its technical derivations are solid. There have been several questions/concerns which are mostly resolved after the rebuttal. As a result, the reviewers have all unanimously recommended accept.

**Award:**

No

---

### Decision · Program_Chairs · 2022-09-14

Accept